# Thyroid hormone regulates distinct paths to maturation in pigment cell lineages

Lauren M Saunders[1,2,3], Abhishek K Mishra[2,3], Andrew J Aman[2,3],
Victor M Lewis[1,2,3], Matthew B Toomey[4†], Jonathan S Packer[1], Xiaojie Qiu[1],
Jose L McFaline-Figueroa[1], Joseph C Corbo[4], Cole Trapnell[1]*, David M Parichy[2,3]*

[1]Department of Genome Sciences, University of Washington, Seattle, United States;
[2]Department of Biology, University of Virginia, Charlottesville, United States;
[3]Department of Cell Biology, University of Virginia, Charlottesville, United States;
[4]Department of Pathology and Immunology, Washington University School of
Medicine, St. Louis, United States

**Abstract** Thyroid hormone (TH) regulates diverse developmental events and can drive disparate cellular outcomes. In zebrafish, TH has opposite effects on neural crest derived pigment cells of the adult stripe pattern, limiting melanophore population expansion, yet increasing yellow/orange xanthophore numbers. To learn how TH elicits seemingly opposite responses in cells having a common embryological origin, we analyzed individual transcriptomes from thousands of neural crest-derived cells, reconstructed developmental trajectories, identified pigment cell-lineage specific responses to TH, and assessed roles for TH receptors. We show that TH promotes maturation of both cell types but in distinct ways. In melanophores, TH drives terminal differentiation, limiting final cell numbers. In xanthophores, TH promotes accumulation of orange carotenoids, making the cells visible. TH receptors act primarily to repress these programs when TH is limiting. Our findings show how a single endocrine factor integrates very different cellular activities during the generation of adult form.
DOI: https://doi.org/10.7554/eLife.45181.001

*For correspondence:
coletrap@uw.edu (CT);
dparichy@virginia.edu (DMP)

Present address: †Department
of Biological Science, University
of Tulsa, Tulsa, United States

Competing interests: The
authors declare that no
competing interests exist.

Reviewing editor: Richard M
White, Memorial Sloan Kettering
Cancer Center, United States

## Introduction

Mechanisms that synchronize developmental signals and integrate them across cell types and organ systems remain poorly defined but are fundamentally important to both development and evolution of adult form (*Atchley and Hall, 1991*; *Ebisuya and Briscoe, 2018*). A powerful system for elucidating how organisms coordinate fate specification and differentiation with morphogenesis is the array of cell types that arise from embryonic neural crest (NC), a key innovation of vertebrates (*Gans and Northcutt, 1983*). NC cells disperse throughout the body, contributing peripheral neurons and glia, osteoblasts and chondrocytes, pigment cells and other derivatives. Differences in the patterning of these cells underlie much of vertebrate diversification.

Thyroid hormone (TH) coordinates post-embryonic development of NC and other derivatives through mechanisms that are incompletely characterized (*Brent, 2012*; *Brown and Cai, 2007*; *Sachs and Buchholz, 2017*; *Shi, 1999*). During the abrupt metamorphosis of amphibians, TH drives outcomes as disparate as tail resorption and limb outgrowth. In the more protracted post-embryonic development of zebrafish—which has similarities to fetal and neonatal development of mammals (*Parichy et al., 2009*)—TH coordinates modifications to several traits including pigmentation. Remarkably, TH has seemingly opposite effects on two classes of NC-derived pigment cells, curtailing the population of black melanophores yet promoting development of yellow/orange xanthophores; fish lacking TH have about twice the normal number of melanophores and lack visible xanthophores (*Figure 1A*) (*McMenamin et al., 2014*).

**eLife digest** Hormones control the development of animals from embryos all the way into adulthood. For example, thyroid hormone is needed to transform a tadpole into an adult frog, and it is essential for developing the nervous system and regulating metabolism in countless other adult animals. However, it remains unclear how a single hormone can control such a diverse range of outcomes.

One way to learn more about the effects of thyroid hormone during development is to study zebrafish pigmentation. Pigment cells arise from a group of stem cells in the embryo called the neural crest. Two of these pigment cells respond to thyroid hormone in different ways: it causes orange pigment cells called xanthophores to expand in number, and at the same time limits the number of black pigment cells called melanophores.

To investigate how thyroid hormone effects the numbers of these pigment cells Saunders et al. mapped the active genes of individual cells derived from the neural crest. Further experiments were then performed on the fish themselves based on these gene activity maps. Comparing fish with and without thyroid hormone showed the hormone actually helps both orange and black pigment cells to mature, but in very different ways. For the orange xanthophores, thyroid hormone drives cells already poised to change into their adult form to acquire orange pigments. For the black melanophores, it causes them to mature into their final non-dividing adult state. This results in xanthophores becoming visible just as the number of melanophores is forced to curtail. Saunders et al. also found the receptor for thyroid hormone acts like a brake for both pigment cells, making sure neither cell type matures in the absence of the hormone.

These experiments show how one hormone can independently regulate different cell types as they mature into their adult form. The finding that thyroid hormone limits the growth of melanocytes may explain why people who produce too little thyroid hormone are at greater risk of melanoma – a form of skin cancer that starts in the melanocytes. But more studies are needed to see if thyroid hormone has the same limiting effect on melanocytes in humans.

DOI: https://doi.org/10.7554/eLife.45181.002

We asked how a single endocrine factor can have such different effects on cells sharing a common embryonic origin. Using transcriptomic analyses of individual cell states, we comprehensively defined the context for TH activities by identifying populations and subpopulations of adult NC derivatives. We then assessed the consequences of TH status for lineage maturation across pigment cell classes. Our analyses showed that TH drives maturation of cells committed to melanophore and xanthophore fates through different mechanisms, promoting terminal differentiation and proliferative arrest in melanophores, and carotenoid-dependent repigmentation in xanthophores. These mechanisms reflect different developmental histories of melanophores and xanthophores and yield different cell-type abundances when TH is absent. Our findings provide insights into post-embryonic NC lineages, contribute resources for studying adult pigment cells and other NC-derived cell types, and illustrate how a circulating endocrine factor influences local cell behaviors to coordinate adult trait development.

## Results

### Post-embryonic NC-derived subpopulations revealed by single-cell RNA sequencing

To explain the pigment cell imbalance of hypothyroid fish, we envisaged two models for TH activity during normal development (*Figure 1B*). In the first model, TH influences states of specification, directing multipotent cells away from one fate and toward the other, or preventing the transdifferentiation of cells already committed to a particular fate. In the second model, TH influences cells that are already committed and remain committed, to their fates. In this scenario, discordant effects across lineages might be observed if TH promotes a cellular process in one lineage that amplifies its population, while simultaneously inhibiting the same process in the other lineage to restrain its population.

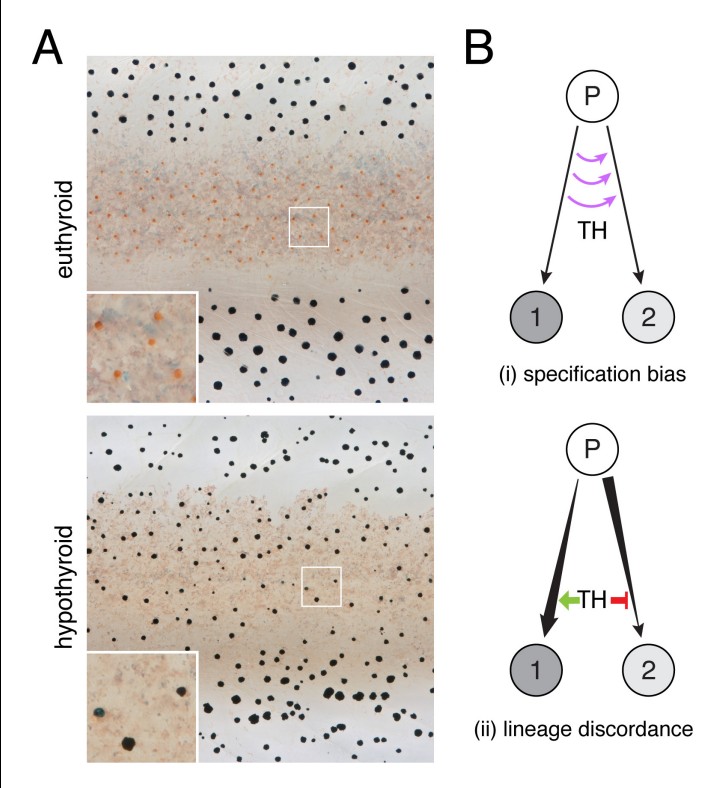

**Figure 1.** TH-dependent phenotypes and models for TH action. (**A**) Euthyroid and hypothyroid zebrafish [stage 10 (mm) standardized standard length (SSL) (*Parichy et al., 2009*); ~21 d post-fertilization, dpf]. Insets, yellow/orange xanthophores of euthyroid fish and absence of these cells in hypothyroid fish. (**B**) Models for TH effects on alternative cell types derived from a common progenitor (**P**), by regulating: (**i**) cell fate specification; or (**ii**) amplification and restraint of committed cell-types by differential effects on morphogenesis or differentiation.
DOI: https://doi.org/10.7554/eLife.45181.003

To evaluate the applicability of these models to TH-dependent regulation of pigment cell populations, we sought to capture the range of intermediate states through which these cells transit during normal and hypothyroid development. Accordingly, we sequenced transcriptomes of thousands of individual NC-derived cells isolated from trunks of euthyroid and hypothyroid fish (*Figure 2—figure supplements 1* and *2*). Dimensionality reduction (*Becht et al., 2018*; *Cao et al., 2019*) followed by unsupervised clustering identified melanophores, xanthophores and a third class of NC-derived pigment cells, iridescent iridophores (*Figure 2A and B*; *Supplementary file 1*). A cluster likely corresponding to multipotent pigment cell progenitors (*Budi et al., 2011*; *Singh et al., 2016*) was marked by genes encoding pigment cell transcription factors, general markers of mesenchymal NC and factors associated with proliferation and migration but not pigment synthesis (*Figure 2—figure supplement 3*; *Supplementary file 2—Table 1*). Some cells within this cluster also expressed the zebrafish-specific embryonic NC marker *crestin*, which is generally down-regulated at later developmental stages but is still expressed in a subset of presumptive progenitor cells (*Budi et al., 2011*).

Beyond pigment cells and their presumptive precursors, other clusters were identifiable as neurons, Schwann cells, other glia, and chromaffin cells. An additional cluster expressed markers suggestive of proliferative, non-pigmentary progenitors, and one large cluster ('unknown') was not readily assignable to NC-derived populations described previously. Bioinformatic comparisons across all clusters revealed distinct expression profiles of genes encoding ligands and receptors, cell adhesion molecules, and products likely to have diverged in function after the teleost-specific whole-genome duplication (*Figure 2—figure supplement 4*).

The larva-to-adult transformation of zebrafish entails changes in a variety of traits including NC derivatives (*Parichy et al., 2009*). In some instances, cell types at different stages that are

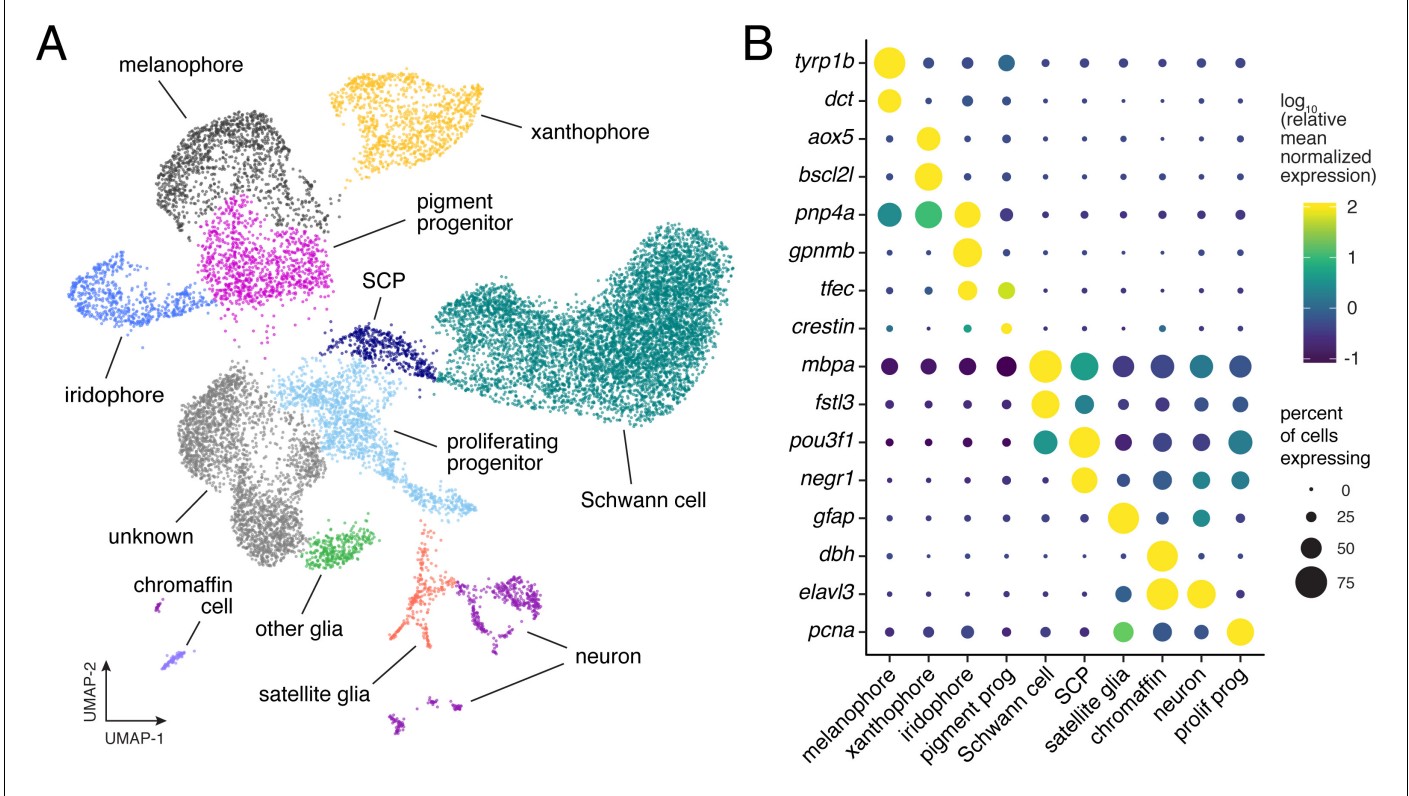

**Figure 2.** Single-cell transcriptomic identification of post-embryonic NC-derived cell types. (**A**) Cell-type assignments for clusters of cells (*n* = 16,150) from euthyroid and hypothyroid fish. Cell types known to be of non-NC derivation are not shown (***Figure 2—figure supplement 2D and E***). (**B**) Known cell-type marker genes and new candidate markers (for cluster-specific genes, see ***Supplementary file 2—Table 1***).
DOI: https://doi.org/10.7554/eLife.45181.004
The following figure supplements are available for figure 2:

**Figure supplement 1.** Experimental design and isolation of NC-derived cells from post-embryonic zebrafish.
DOI: https://doi.org/10.7554/eLife.45181.005
**Figure supplement 2.** Population characteristics for full scRNA-Seq dataset from post-embryonic zebrafish.
DOI: https://doi.org/10.7554/eLife.45181.006
**Figure supplement 3.** Genes enriched in pigment progenitor clusters include known markers of embryonic NC cells.
DOI: https://doi.org/10.7554/eLife.45181.007
**Figure supplement 4.** Distinct domains of gene expression across diverse NC derivatives.
DOI: https://doi.org/10.7554/eLife.45181.008
**Figure supplement 5.** Similarities and differences between EL and adult gene expression programs.
DOI: https://doi.org/10.7554/eLife.45181.009

superficially similar (e.g. larval vs. adult melanophores) can be distinguished by different genetic requirements (***Budi et al., 2008***; ***Larson et al., 2010***; ***Parichy et al., 1999***), raising the possibility that distinct gene expression programs regulate early larval and adult populations. If so, we predicted that NC-derived cells isolated from middle larval–juvenile stages, during development of the adult phenotype (i.e. ***Figure 2a***), should form clusters distinct from cells that developed during embryonic stages to form the embryonic–early larval ('EL') phenotype. To test this idea, we isolated EL NC-derived cells, which clustered in identifiable cell types similar to those of middle-larval juvenile stages (***Figure 2—figure supplement 5A***). Combining profiles for cells at different stages failed to reveal non-contiguous, life-stage-specific clusters, although some EL cells occupied subsets of transcriptomic space relative to their broader cell type (e.g. melanophores) (***Figure 2—figure supplement 5B–D***). These data do not indicate markedly different transcriptomic programs of NC-derived

cell types across life stages, despite the existence of some stage-specific requirements for particular genes and pathways.

Overall, our survey captured numerous NC-derived cell types, including abundant pigment cells and progenitors, and revealed substantial variation in gene expression programs among them.

## Pigment cell sub-classes and gene expression dynamics across differentiation

To understand the gene expression context in which TH impacts each pigment cell type, we compared pigment cells and progenitors, the lineages of which have been described (*Budi et al., 2011*; *Mahalwar et al., 2014*; *McMenamin et al., 2014*; *Patterson and Parichy, 2019*; *Singh et al., 2016*) (*Figure 3A*). These analyses revealed subsets of melanophores and xanthophores (*Figure 3B*), consistent with differences in states of differentiation and morphogenetic behaviors (*Eom et al., 2015*; *Parichy et al., 2000b*; *Parichy and Spiewak, 2015*). For example, cells of subcluster melanophore 2 exhibited low levels of transcriptional activity and expressed fewer genes, suggesting a more advanced state of differentiation, as compared to cells of melanophore 1 (*Figure 3—figure supplement 1*). Likewise cells of xanthophore 1 had fewer transcripts and expressed fewer genes than cells of xanthophore 2, suggesting they may represent undifferentiated, cryptic xanthophores and actively differentiating populations, respectively (*McMenamin et al., 2014*).

Additional surveys of these data revealed new markers of xanthophore and iridophore lineages (*Figure 3—figure supplements 2* and *3*), and cell-type-specific expression of some previously identified markers [e.g. *tyrp1b*, *aox5*, *tfec* (*Lister et al., 2011*; *McMenamin et al., 2014*) (*Figure 3C*). Expression of other genes was broader than might be expected from mutational or other analyses (*Figure 3—figure supplement 4*); for example *mitfa*, encoding a transcription factor required for melanophore fate specification (*Lister et al., 1999*) was expressed in melanophores and progenitors, but also xanthophores (*Figure 3C*), consistent with prior reports (*Eom et al., 2012*; *Parichy et al., 2000b*).

To characterize transcriptional dynamics through lineage maturation, we pseudotemporally ordered cells (*Qiu et al., 2017a*; *Qiu et al., 2017b*; *Trapnell et al., 2014*), yielding a differentiation trajectory with each pigment cell type arising from a common progenitor (*Figure 3D*). This topology differed from known lineage relationships (*Figure 3A*) but was consistent with similarity of EL and mid-larval/juvenile gene expression programs (*Figure 2—figure supplement 5D*). Branch expression analysis modeling (BEAM) (*Qiu et al., 2017a*) confirmed that genes with functions in specification (e.g. *mitfa* in melanophores) were expressed early in pseudotime whereas genes associated with differentiation [e.g. *dct*, encoding a melanin synthesis enzyme (*Kelsh et al., 2000b*)] were expressed late (*Figure 3E*; *Figure 3—figure supplement 5A*). These analyses revealed dynamics of dozens of genes potentially identifying discrete processes in lineage-specific maturation (*Supplementary file 2—Table 2*) as well as broader trends. For example, transcripts per cell declined in melanophores but not iridophores, consistent with an expectation of reduced RNA abundance as melanophores—but not iridophores—exit the cell cycle with maturation (*Figure 3—figure supplement 5B*) (*Budi et al., 2011*; *Darzynkiewicz et al., 1980*; *McMenamin et al., 2014*; *Spiewak et al., 2018*).

## TH-independence of pigment cell fate specification and absence of lineage-specific restraints on developmental progress

Resolution of pigment cell states through their development allowed us to test if TH functions in fate specification (*Figure 1B–i*). If so, the excess melanophores and missing xanthophores of hypothyroid fish should reflect biases on specification of multipotent progenitors, or the transdifferentiation (*Lewis et al., 2019*; *Niu, 1954*) of initially specified cells. Such alterations should be evident in reduced-dimension transcriptomic space as strong skew in the apportionment of cells between branches or abnormal paths in the cellular trajectory, respectively. Yet, euthyroid and hypothyroid trajectories were topologically equivalent. Moreover, pigment cell progenitors were not depleted in hypothyroid fish as might occur were these cells being allocated inappropriately as melanophores (*Figure 4A–D*).

Through a second model—lineage discordance—TH could have opposite effects on cells already committed to particular fates, selectively amplifying one cell type while simultaneously repressing amplification of the other (*Figure 1B–ii*). For example, TH could promote differentiation of

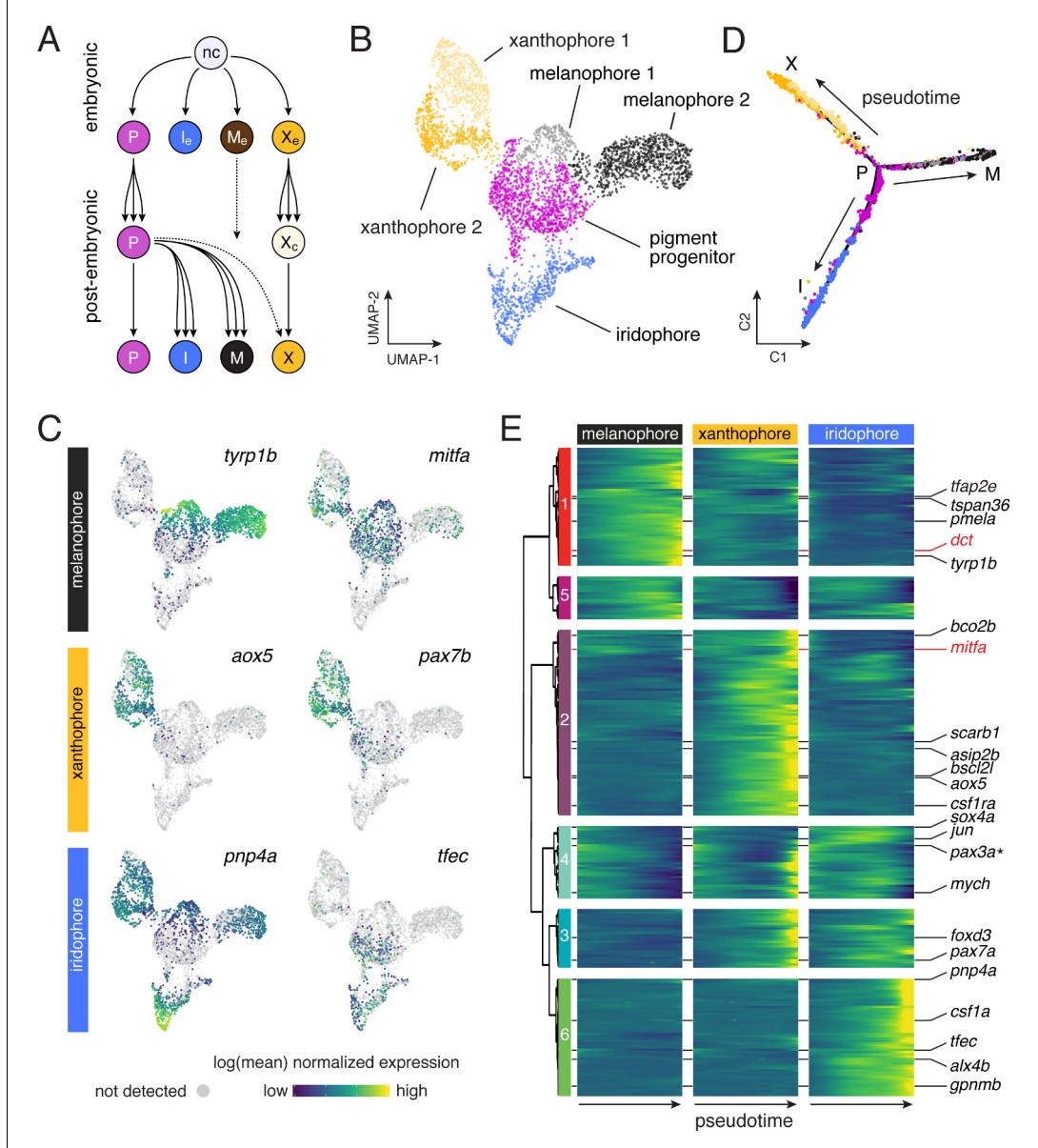

**Figure 3.** Pigment cell subpopulations and dynamics of gene expression across pigment cell lineages. (A) Established lineage relationships of embryonic (e) and post-embryonic pigment cells. Multipotent pigment cell progenitors (P) in the peripheral nervous system generate adult iridophores (I), melanophores (M) and some xanthophores (X). A few embryonic melanophores ($M_e$) persist, whereas embryonic xanthophores ($X_e$) proliferate and lose their pigment to enter a cryptic phase ($X_c$), and then reacquire pigment late in pattern formation to form most adult xanthophores (*McMenamin et al., 2014*). (B) Sub-clusters of melanophores and xanthophores with distinct gene expression signatures. (C) Pigment cell clusters defined by markers for each cell-type. (D–E) Pseudotemporal ordering (D) and BEAM (E) revealed dynamics of gene expression over pseudotime for each pigment cell branch [$q < 6.0E\text{-}11$ for all genes except *pax3a*, starred, $q = 0.03$], expressed as anticipated during early pseudotime in each branch].
DOI: https://doi.org/10.7554/eLife.45181.010

The following figure supplements are available for figure 3:

**Figure supplement 1.** Differences between melanophore and xanthophore sub-populations revealed distinct levels and types of transcriptional activity.
DOI: https://doi.org/10.7554/eLife.45181.011

**Figure supplement 2.** Xanthophore cluster-specific expression identifies novel xanthophore markers.
DOI: https://doi.org/10.7554/eLife.45181.012

**Figure supplement 3.** Iridophore cluster-specific expression identifies novel iridophore markers.
DOI: https://doi.org/10.7554/eLife.45181.013

**Figure supplement 4.** Genes identified as zebrafish pigmentation mutants often had expression domains beyond affected cell types.

*Figure 3 continued on next page*

*Figure 3 continued*

DOI: https://doi.org/10.7554/eLife.45181.014

**Figure supplement 5.** Dynamics of gene expression over pseudotime recapitulated distinct melanophore and iridophore differentiation programs.
DOI: https://doi.org/10.7554/eLife.45181.015

xanthoblasts to xanthophores, but prevent differentiation of melanoblasts to melanophores. Alternatively, TH could be a survival factor in the xanthophore lineage but a pruning factor in the melanophore lineage. Terminal phenotypes of both hypothyroid and hyperthyroid mutant fish are consistent with such a mechanism (*McMenamin et al., 2014*). If TH has discordant effects between lineages, we predicted that hypothyroid fish should exhibit a strong depletion of xanthophores from the end of their branch of the trajectory, whereas melanophores should be strongly over-represented near the tip of their branch. Yet, empirical distributions of pigment cell states in hypothyroid fish were all biased towards earlier steps in pseudotime, sometimes severely (*Figure 4E*). Indeed, prior analyses showed that addition of exogenous TH to hypothyroid cells ex vivo can promote differentiation of unpigmented melanoblasts to melanophores (*McMenamin et al., 2014*), contrary to the idea that TH specifically blocks melanophore development. Together these findings allow us to reject a model in which TH regulation of pigment cell abundance in the adult fish occurs through discordant effects on specific cellular processes across lineages.

## TH promotes a melanophore maturation program

Having rejected both of our initial models (*Figure 1B*), we considered a third possibility, that TH promotes the maturation of both lineages, but in distinct ways. For melanophores, inspection of transcriptomic states and cellular phenotypes supported a role for TH in promoting maturation of this lineage. Genes expressed during terminal differentiation of melanophores from euthyroid fish (e.g. *tfap2a*, *tyrp1b*) were expressed at lower levels in melanophores of hypothyroid fish, suggesting an impediment to maturation in the absence of TH (*Figure 5A*).

To test further test the idea that TH promotes the maturation of melanophores, we examined additional cellular phenotypes. Melanophores of juvenile euthyroid fish tended to be uniformly well-melanized and stellate, whereas melanophores of juvenile hypothyroid fish were variably melanized and dendritic (*Figure 5B*), reminiscent of earlier stages of melanophore development in wild-type (*Eom et al., 2012*; *Parichy and Turner, 2003*). Quantification of melanin content within individual cells confirmed that melanophores of euthyroid fish are more heavily melanized than those of hypothyroid fish (*Figure 5C*).

Prior analyses indicated that melanophores of euthyroid fish fail to divide whereas those of hypothyroid fish continue to do so (*McMenamin et al., 2014*). These findings raised the possibility that melanophores of euthyroid fish might exhibit signs of cellular senescence or other indications of proliferative cessation not observed in melanophores of hypothyroid fish. Human nevus melanocytes, and melanophores of teleost melanoma models, exhibit senescent or senescent-like phenotypes and can be multinucleated (*Leikam et al., 2015*; *Leikam et al., 2008*; *Regneri et al., 2019*; *Savchenko, 1988*). Accordingly, we asked whether similar attributes were evident for zebrafish stripe melanophores. When plated ex vivo, some stripe melanophores exhibited senescence-associated β-galactosidase (SA-β-gal) activity (*Figure 5—figure supplement 1A*), although we were unable to score such staining reliably, precluding comparisons across TH conditions.

SA-β-gal staining results from lysosomal β-gal activity and both β-gal activity and lysosome number increase in aging cells (*Kurz et al., 2000*; *Lee et al., 2006*). We therefore quantified lysosome-specific Lysotracker labeling (*Figure 5—figure supplement 1B and F*) of melanophores by fluorescence activated cell sorting *tyrp1b*:palm-mCherry+ melanophores. Lysosomal contents of melanophores from euthyroid fish were greater than melanophores from hypothyroid fish (*Figure 5—figure supplement 1C*). Measurements of forward scatter (FSC-A) also suggested that melanophores from juvenile euthyroid fish were larger than melanophores from hypothyroid fish (*Figure 5—figure supplement 1D*), although FSC-A can be influenced by cell-size-independent factors as well (*Tzur et al., 2011*).

Finally we examined multinucleation, a condition linked to increased cell survival and size (*Orr-Weaver, 2015*; *Usui et al., 2018*). In euthyroid fish, ~20% of melanophores were binucleate near the

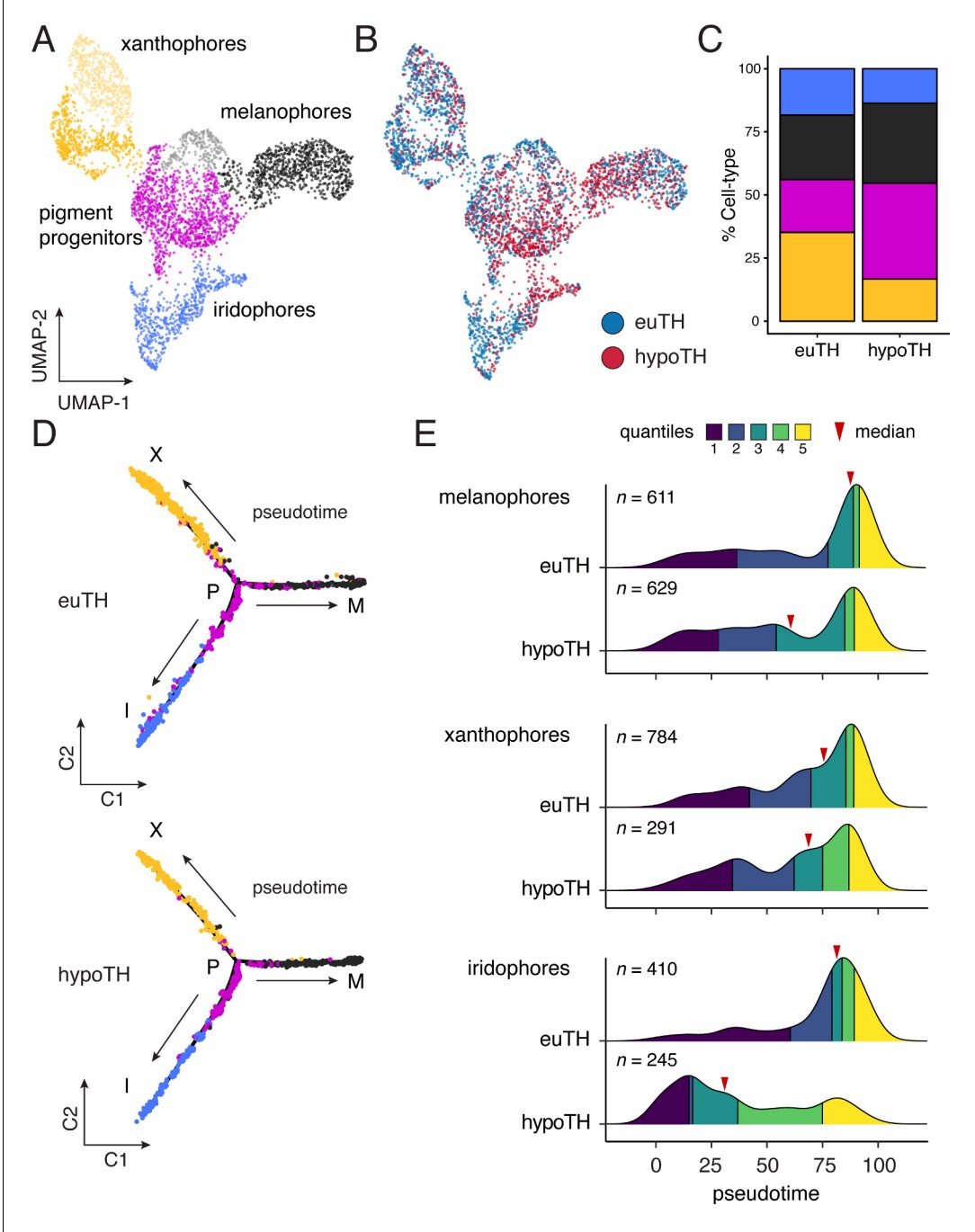

**Figure 4.** TH biased pigment cell lineages toward later steps of pseudotime. (A) UMAP dimensionality reduction of euthyroid and hypothyroid pigment cells and pigment progenitors. Sub-clustering revealed two xanthophore and two melanophore clusters (indicated by different shades of yellow and gray, respectively). (B) Pigment cells colored by TH status. Euthyroid and hypothyroid cells were generally intermixed with some biases apparent within melanophore and iridophore clusters. (C) Percentages of each pigment cell class by TH-status. Colors are consistent with other pigment cell plots. Of cells captured, a higher proportion of pigment cells from euthyroid fish were xanthophores and iridophores compared to those from hypothyroid fish. (D) Trajectories for euthyroid and hypothyroid pigment cells. Broad differences in trajectory topologies were not apparent between the two conditions. (E) Distributions of each pigment cell-type across pseudotime by condition. For each trajectory branch (mel, xan, irid), hypoTH cells were biased toward early pseudotime (Wilcoxon signed-rank tests, mel: Z = −6.54, p<0.0001, xan: Z = −4.54, p<0.0001, irid: Z = −13.55, p<0.0001). Median is indicated by red arrowhead and different colors demarcate quartiles over pseudotime.

DOI: https://doi.org/10.7554/eLife.45181.016

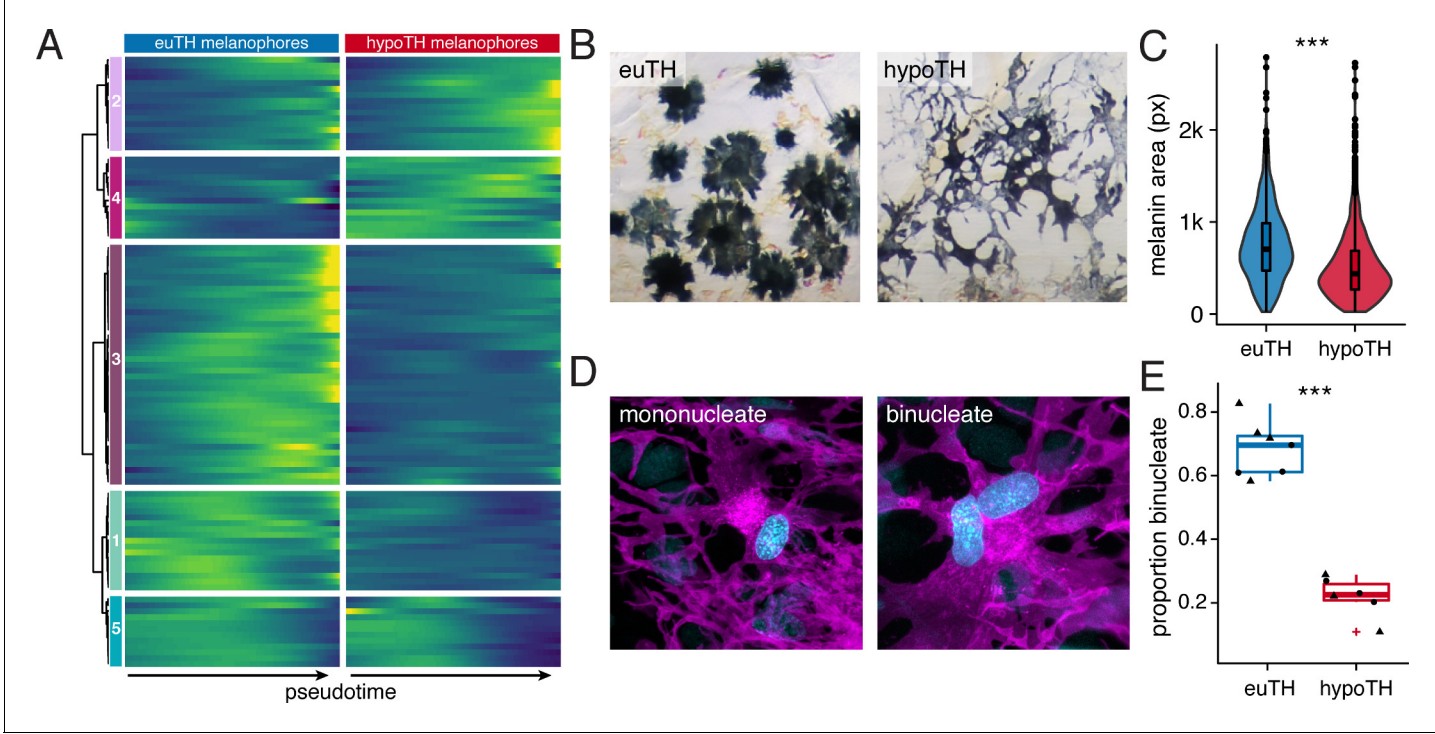

**Figure 5.** TH promoted melanophore maturation by measures of transcriptomic state and cellular phenotype. (**A**) Gene expression differences between melanophores over pseudotime by TH-status ($q < 1E-7$, genes expressed in >10% of melanophores). Heatmap is hierarchically clustered by row (method, Ward D2). The largest cluster (#3) contains 41% of the genes and represents loci expressed late in pseudotime of euthyroid melanophores but downregulated in hypothyroid melanophores (e.g. *tfap2a* and *tyrp1b*), identifying novel candidate genes for roles in melanophore maturation (see *Supplementary file 2—Table 6*, in which published melanophore-related genes are highlighted) (*Baxter et al., 2019*). (**B**) Euthyroid melanophores tended to be highly melanized and stellate, whereas hypothyroid melanophores were variably melanized and often dendritic. (**C**) Quantification of melanin contents per cell, as estimated by area of pixels (px) having melanin following contraction of melanin granules in response to epinephrine (e.g. *Figure 1A*). Melanophores of euthyroid fish contained more melanin than those of hypothyroid fish ($F_{1,2710}$=271.2, p<0.0001), after controlling for individual variation among fish within TH conditions ($F_{10,2710}$=8.5, p<0.0001; sample sizes: *n* = 1180 cells from five euthyroid fish, *n* = 1542 cells from five hypothyroid fish). If planar areas of concentrated melanin granules are assumed spherical, then euthyroid melanophores had on average ~1.7 x the total melanin content of hypothyroid melanophores. (Data in supplementary file *Figure 5—source data 1*) (**D**) Fully differentiated melanophores of zebrafish are often binucleate (*Usui et al., 2018*). Left panel shows a binucleate stripe melanophore in a euthyroid fish (12 SSL). Right panel shows a mononucleate melanophore in a hypothyroid fish at the same stage. Magenta, membrane labeling of melanophores by *tyrp1b:palm-mCherry*. Blue, nuclei revealed by *tuba8l3:nEosFP*. (**E**) Euthyroid fish had proportionally more binucleate melanophores than hypothyroid fish ($\chi^2$=230.3, d.f. = 1, p<0.0001) after controlling for a higher incidence of binucleation in developmentally more advanced fish overall (11.5–13 SSL; $\chi^2$ = 5.5, d.f. = 1, p<0.05). Individual points indicate proportions of binucleate melanophores observed in dorsal stripes (circles) and ventral stripes (diamonds), which did not differ significantly (p=0.8; sample sizes: *n* = 383 melanophores in four euthyroid fish, *n* = 706 melanophores in three hypothyroid fish). (Data in supplementary file *Figure 5—source data 2*).

DOI: https://doi.org/10.7554/eLife.45181.017

The following source data and figure supplements are available for figure 5:

**Source data 1.** Melanin content data corresponding to *Figure 5C*.
DOI: https://doi.org/10.7554/eLife.45181.021
**Source data 2.** Melanophore binuclation incidence data corresponding to *Figure 5E*.
DOI: https://doi.org/10.7554/eLife.45181.022
**Figure supplement 1.** Metrics of melanophore maturation in response to TH.
DOI: https://doi.org/10.7554/eLife.45181.018
**Figure supplement 1—source data 1.** Lysotracker FACS data corresponding to *Figure 5—figure supplement 1C and D*.
DOI: https://doi.org/10.7554/eLife.45181.019
**Figure supplement 1—source data 2.** Melanophore binuclation incidence data corresponding to *Figure 5—figure supplement 1E*.
DOI: https://doi.org/10.7554/eLife.45181.020

onset of adult melanophore differentiation but >50% were binucleate by juvenile stages, confirming an overall increase in binucleation with somatic stage and melanophore age (*Figure 5—figure supplement 1E*). In stage-matched comparisons for TH status, ~70% of melanophores from euthyroid fish were binucleated, whereas only ~25% of melanophores from hypothyroid fish were in this state (*Figure 5D and E*).

Collectively, our observations and those of *McMenamin et al. (2014)* suggest a model in which TH drives melanophores into a terminally differentiated state of increased melanization, larger size and lysosomal content, binucleation, and proliferative cessation.

## TH promotes carotenoid-dependent xanthophore re-pigmentation during adult development

We next examined TH functions specific to the xanthophore lineage. Most adult xanthophores develop directly from EL xanthophores that lose their pigment and then reacquire it late in adult pattern formation (*Figure 3A*) (*McMenamin et al., 2014*). Because xanthophores of hypothyroid fish persist, albeit in a cryptic state, we predicted that TH effects should be less pervasive in these cells than in melanophores that develop de novo from transit amplifying cells originating from multipotent progenitors. Indeed, fewer genes were expressed differentially between TH backgrounds in xanthophore than melanophore lineages (3.6% vs. 9%; *Figure 6A*). Prominent among these were several loci implicated in, or plausibly associated with, the processing of yellow/orange carotenoids (*Figure 6B and C*; *Figure 6—figure supplement 1*), dietarily derived pigments that contribute to xanthophore coloration (*Schartl et al., 2016*; *Toews et al., 2017*).

Differences in carotenoid gene expression suggested a corresponding pigmentation deficiency in xanthophores of hypothyroid fish that we confirmed by HPLC, histology, and transmission electron microscopy (*Figure 6D*; *Figure 6—figure supplement 2*). Among carotenoid genes, *scavenger receptor B1* (*scarb1*) encodes a high-density lipoprotein receptor essential for carotenoid accumulation in birds and invertebrates (*Kiefer et al., 2002*; *Toomey et al., 2017*) and we found it to be required in zebrafish for carotenoid deposition, although not cell persistence (*Figure 6—figure supplement 3A and B*). *scarb1* was expressed more highly in xanthophores of euthyroid than hypothyroid fish ($q$ = 1.1E-10) (*Figure 6B and E*; *Figure 6—figure supplement 1*) and exogenous TH was sufficient to rescue both expression and carotenoid deposition (*Figure 6F*; *Figure 6—figure supplement 3C*). Together these findings demonstrate an essential role for TH in carotenoid pigmentation and suggest that TH modulation of a suite of carotenoid pathway genes is required for cryptic xanthophores to re-pigment during adult pattern formation.

The distinct phases of xanthophore EL and adult pigmentation (*McMenamin et al., 2014*), and the TH-dependence of the latter, led us to ask whether mechanisms underlying coloration might be stage-specific. In contrast to the defect of adult xanthophore pigmentation in *scarb1* mutants, we found that 5 dpf larval xanthophores were indistinguishable from wild-type (*Figure 6—figure supplement 4A*). Conversely, mutants lacking xanthophore pigmentation at 5 dpf have normal adult xanthophores (*Lister, 2019*; *Odenthal et al., 1996*). Because two pigment classes—carotenoids and pteridines—can contribute to xanthophore coloration, we hypothesized that visible colors at different stages depend on different pathways. Carotenoids were undetectable in euthyroid 5 dpf larvae, and carotenoid-related genes were expressed at lower levels in EL xanthophores than adult xanthophores (*Figure 6—figure supplement 4B and C*). By contrast, pteridine pathway genes tended to be expressed similarly across stages regardless of TH status and were even moderately upregulated in hypothyroid xanthophores (*Figure 6C*, *Figure 6—figure supplement 4C*). Pteridine autofluorescence and pterinosomes were also indistinguishable between euthyroid and hypothyroid fish (*Figure 6—figure supplement 4D*; *Figure 6—figure supplement 2B*) despite the overt difference in xanthophore color with TH status (*Figure 1A*; *McMenamin et al., 2014*). Together, these observations imply that TH induces new, carotenoid-based pigmentation, allowing transiently cryptic xanthophores to reacquire coloration during adult pattern development. TH therefore drives maturation of both xanthophores and melanophores yet has markedly different roles in each lineage.

## Adult pigment cell maturation programs are gated by TH receptors

Finally, to understand how TH effects are transduced in pigment cell lineages, we evaluated roles for TH nuclear receptors (TRs) that classically activate target genes when ligand (T3) is present but

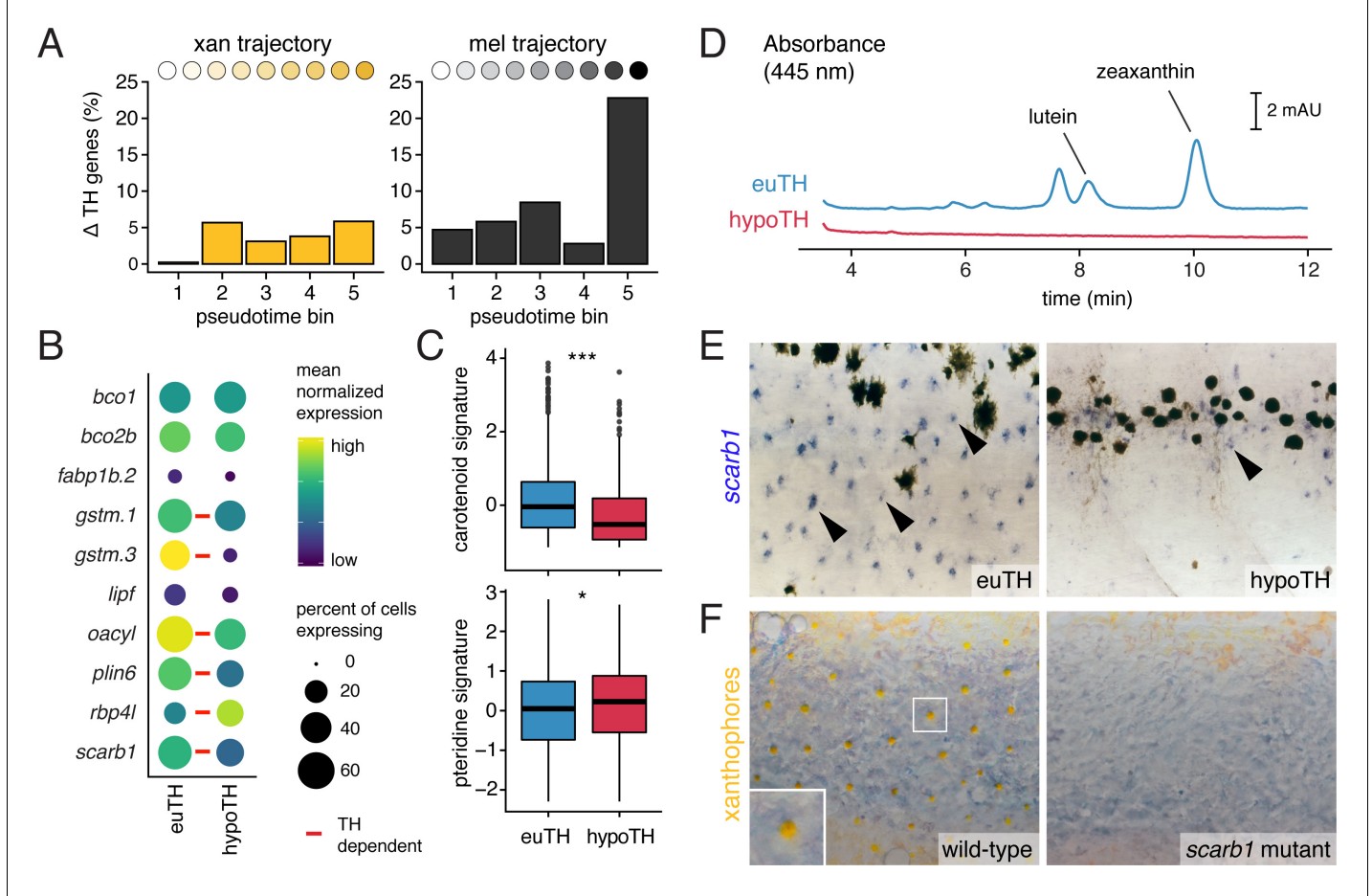

**Figure 6.** TH promotes xanthophore maturation via *scarb1*-dependent carotenoid uptake. (A) Proportions of differentially expressed genes in euthyroid and hypothyroid cells across pseudotime bins. Xanthophores expressed fewer TH-dependent genes than melanophores (expressed gene cutoff = 2% of bin expressing, DEGs are genes with *q* < 0.05 and fold change >1.5X). Of 160 xanthophore DEGs and 519 melanophore DEGs, only 58 were found to be overlapping. (B) TH-dependent expression of genes related to carotenoid pigmentation in xanthophores. Red bars: *q* < 0.05, log$_2$ fold-change ≥2.0. (C) Carotenoid pathway gene expression score was higher in xanthophore lineage cells of euthyroid fish compared to hypothyroid fish (p=1.5E-15, Wilcoxon). By contrast, pteridine pathway gene expression was marginally lower in cells from euthyroid fish (p=0.01). Box-and-whisker plots represent scores across groups (center line, median; box limits, upper and lower quartiles; whiskers, 1.5x interquartile range; points, outliers). (D) Carotenoids were detected by HPLC in skin containing xanthophores of euthyroid but not hypothyroid fish (11 SSL). (E) *scarb1* expression in euthyroid and hypothyroid zebrafish (10 SSL). (F) *scarb1* mutants lacked mature, yellow xanthophores (12 SSL).

DOI: https://doi.org/10.7554/eLife.45181.023

The following source data and figure supplements are available for figure 6:

**Figure supplement 1.** Expression of multiple carotenoid-related genes in xanthophores are affected by TH.
DOI: https://doi.org/10.7554/eLife.45181.024
**Figure supplement 1—source data 1.** Xanthophore lipid droplet incidence corresponding to *Figure 6—figure supplement 2A*.
DOI: https://doi.org/10.7554/eLife.45181.025
**Figure supplement 2.** TH promotes development of lipid-filled carotenoid droplets in xanthophores.
DOI: https://doi.org/10.7554/eLife.45181.026
**Figure supplement 3.** *scarb1* is specifically involved in xanthophore maturation and is induced by TH.
DOI: https://doi.org/10.7554/eLife.45181.027
**Figure supplement 4.** Xanthophores switch yellow pigmentation programs during the larval-to-adult transition.
DOI: https://doi.org/10.7554/eLife.45181.028

repress gene expression when ligand is absent (*Brent, 2012*; *Buchholz et al., 2003*; *Hörlein et al., 1995*). Genes encoding each of the three zebrafish TRs (*thraa, thrab, thrb*) were expressed by melanophores and xanthophores, yet presumptive null alleles for each unexpectedly had pigment cell complements and patterns that resembled the wild type (*Figure 7A*; *Figure 7—figure supplement 1A–D*).

Given the absence of grossly apparent phenotypes for TR mutants, we hypothesized that instead of acting to promote maturation when T3 is present, TRs may function primarily to repress maturation when T3 is limiting. If so, we predicted that xanthophore development in hypothyroid fish should be rescued by mutation of TR. We therefore generated fish lacking TH and TRs. Loss of *thrab*, on its own or in conjunction with loss of *thraa*, partially restored the deposition of carotenoids in interstripe xanthophoes; mutation of all three receptors fully rescued the number of carotenoid-containing xanthophores (*Figure 7B and C*; *Figure 7—figure supplement 1E and F*). TR receptor

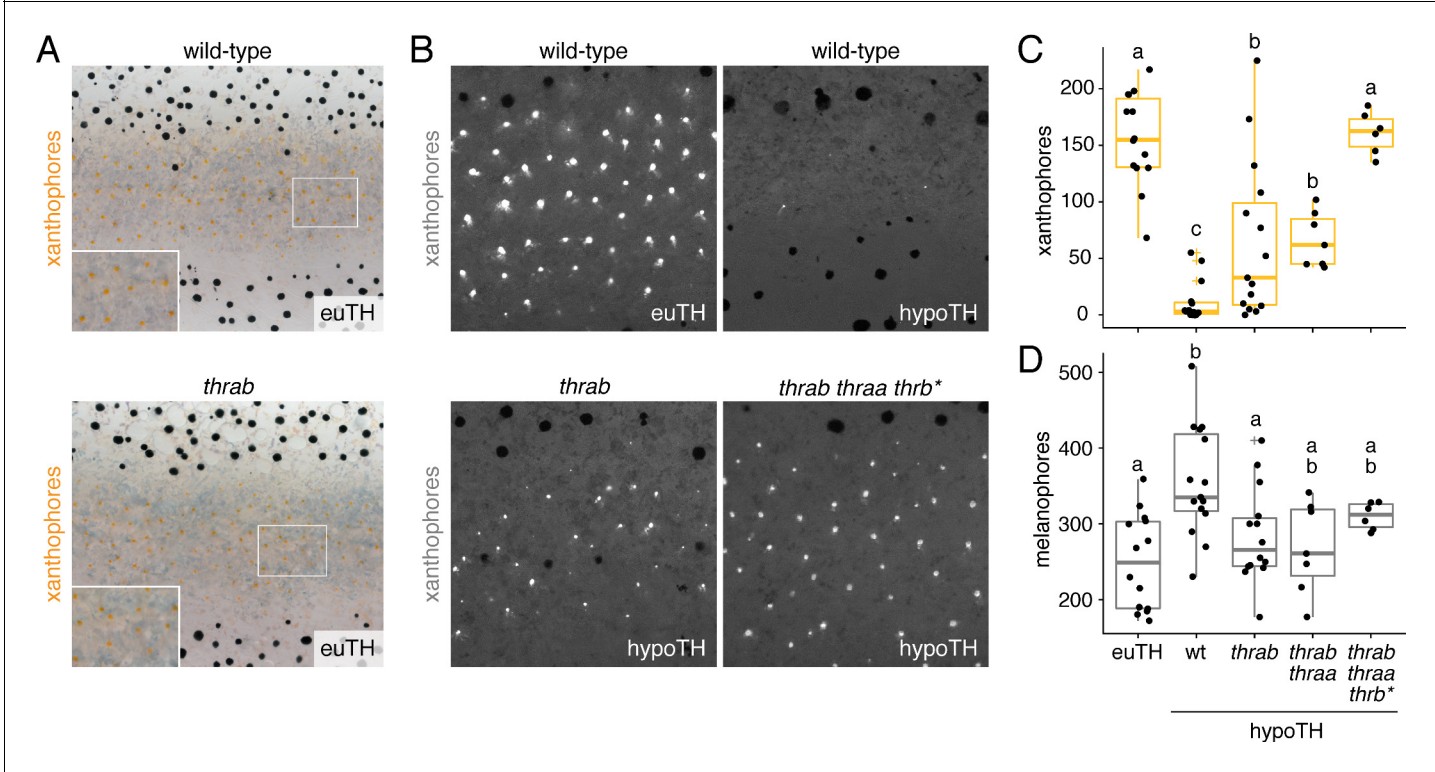

**Figure 7.** TH receptors repress developmental progression of pigment cell lineages. (A) In euthyroid fish, homozygous TR mutants singly and in combination resembled wild-type; shown is *thrab*. (B) Euthyroid fish wild-type for TRs exhibited numerous autofluorescing, carotenoid-containing xanthophores (upper left), whereas hypothyroid fish wild-type for TRs lacked nearly all these cells (upper right). By contrast, hypothyroid fish mutant for TRs developed substantial complements of these cells. Shown here are representative individuals homozygous for *thrab* mutation (lower left) and homozygous *thrab* individuals with somatically induced mutations for *thraa* (*) as well as doubly *thraa* and *thrab* individuals with somatically induced mutations for *thrb* (*thrb**; lower right). Fish are 11.5 SSL. (C, D) Homozygous *thrab* mutation partially rescued numbers of pigmented xanthophores and more fully rescued numbers of melanophores in hypothyroid fish. Somatic mutagenesis of *thraa* in fish homozygous mutant for *thrab* mutants (*thrab thraa**) did not significantly enhance the rescue of xanthophore maturation or melanophore numbers. By contrast somatic mutagenesis of *thrb* in fish doubly homozygous mutant for *thraa* and *thrab* (*thrab thraa thrb**) rescued xanthophore maturation to wild-type levels in the absence of TH. Numbers of visible xanthophores and melanophores were not distinguishable between euthyroid fish wild-type or homozygous mutant for TR mutations either singly or in combination (p>0.1) and are shown combined here. Box plots as in *Figure 6C* with different letters above data indicating significant differences in *post hoc* comparisons (Tukey HSD, p<0.05). (Cell counts in supplementary file *Figure 7—source data 1*.).

DOI: https://doi.org/10.7554/eLife.45181.029

The following source data and figure supplement are available for figure 7:

**Source data 1.** Counts for xanthophores and melanophores corresponding to *Figure 7C and D*.
DOI: https://doi.org/10.7554/eLife.45181.031

**Figure supplement 1.** Zebrafish TR gene expression and mutants.
DOI: https://doi.org/10.7554/eLife.45181.030

mutations likewise reduced the total numbers of melanophores in hypothyroid fish to levels indistinguishable from euthyroid fish (*Figure 7D*).

These findings suggest that repression by unliganded TRs contributes to pigment-associated phenotypes in hypothyroid fish, implying a function for TRs in repressing the repigmentation of xanthophores and terminal differentiation of melanophores until late stages in adult pigment pattern development. Nevertheless, roles for TRs are likely to be complex and outcomes of derepression dependent on context. For example, the simplest model of TR gating would predict that loss of TRs in euthyroid fish should result in the precocious maturation of pigment cells. Yet, we found no evidence for early pigmentation of xanthophores in euthyroid fish homozygous for *thrab* mutation (*Figure 7—figure supplement 1G*), suggesting essential roles for other factors present only at later stages (*Patterson and Parichy, 2013*).

## Discussion

Our study provides insights into how TH coordinates local cellular events during the development of adult form. The stripes of adult zebrafish comprise three major classes of pigment cells that develop at specific stages and from distinct NC sublineages. Perturbations that affect the times of appearance, states of differentiation or morphogenetic behaviors of these cells can dramatically alter pattern by affecting total numbers of cells and the cascade of interactions normally required for spatial organization (*Parichy and Spiewak, 2015*; *Patterson et al., 2014*; *Watanabe and Kondo, 2015*). Fish lacking TH have gross defects in pigment cell numbers and pattern with ~two fold the normal complement of melanophores and the simultaneous absence of visible xanthophores (*McMenamin et al., 2014*). We show that this phenotype arises not because TH normally biases cell fate specification or has discordant effects on a particular cellular behavior that amplifies one cell type while repressing the other. Rather, our findings—combining discovery-based analyses of single-cell transcriptomic states with experiments to test specific cellular hypotheses—suggest a model whereby TH promotes maturation of both melanophores and xanthophores in distinct ways that reflect the developmental histories of these cells (*Figure 8*). Our study provides a glimpse into the diversity of cell states among post-embryonic NC-derivatives and illustrates how a single endocrine factor coordinates diverse cellular behaviors in a complex developmental process.

By sampling individual cell transcriptomes across NC-derived lineages, our study complements prior investigations of lineage relationships, morphogenetic behaviors, genetic requirements, and spatial and cell-type-specific gene expression profiles (*Eom et al., 2015*; *Irion et al., 2016*; *Johnson et al., 1995*; *Kelsh et al., 2017*; *McMenamin et al., 2014*; *Parichy and Spiewak, 2015*; *Singh et al., 2016*; *Singh et al., 2014*). Multipotent progenitors that give rise to adult melanophores, some xanthophores, and iridophores are established in the embryo and reside within peripheral nerves as development progresses (*Budi et al., 2011*; *Budi et al., 2008*; *Camargo-Sosa et al., 2019*; *Dooley et al., 2013a*; *Singh et al., 2016*). As the adult pattern forms, some of these cells migrate to the hypodermis where they differentiate and integrate into dark stripes or light interstripes. The peripheral-nerve association of pigment cell progenitors in zebrafish is reminiscent of nerve-associated Schwann cell precursors that contribute to melanocytes of mammals and birds (*Adameyko et al., 2009*). Our collected cell-types, which include immature and mature glia, differentiating pigment cells, and presumptive progenitors of different types identify new candidate genes for promoting—and recognizing—distinct states of differentiation and morphogenetic activities, and will enable efforts to define how multipotent NC progenitors are maintained and recruited into particular lineages. That corresponding populations of embryonic and adult populations had largely overlapping transcriptomic states additionally highlights the intriguing problem of how specific pathways are deployed reiteratively across life cycle phases to achieve specific morphogenetic outcomes.

Our identification of a role for TH in the adult melanophore lineage illuminates how these cells develop normally and mechanisms that likely contribute to the supernumerary melanophores of hypothyroid fish. Melanoblasts derived from peripheral-nerve associated progenitors are proliferative during adult pigment pattern formation yet this activity largely ceases as the cells differentiate (*Budi et al., 2011*; *McMenamin et al., 2014*). Several lines of evidence suggest that TH promotes melanophore maturation to a terminally differentiated state: in the presence of TH, melanophores were more heavily melanized, larger, had greater lysosomal contents, and were more likely to be

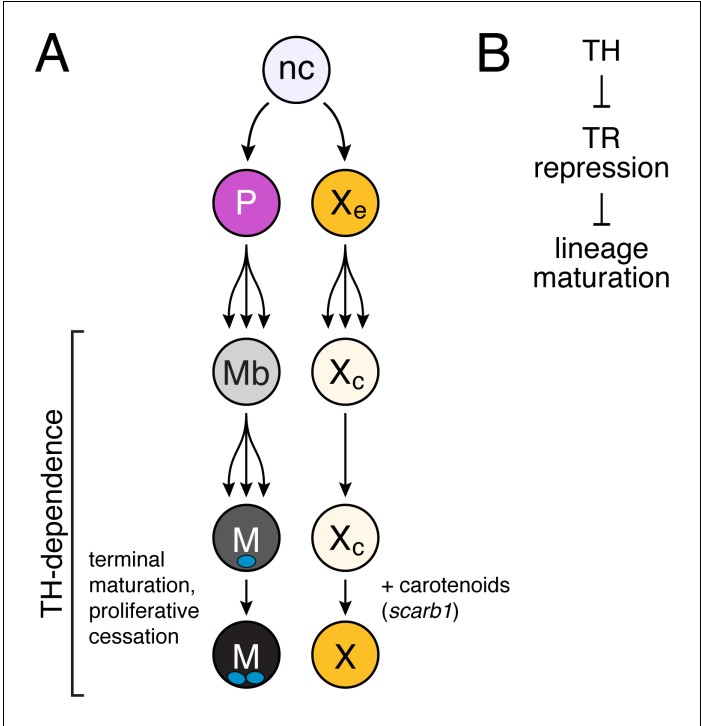

**Figure 8.** Model of TH dependence in zebrafish pigment cell lineages. (**A**) Post-embryonic progenitor-derived, specified adult melanoblasts (Mb) that expand their population and differentiate to a state of proliferative cessation (*McMenamin et al., 2014*), binucleate state, and EL-derived cryptic xanthophores that redifferentiate as carotenoid-containing yellow/orange adult xanthophores. Disparate cell-type-specific outcomes in fish lacking TH reflect differences in events required for maturation between sublineages. (**B**) TH-dependent lineage maturation involves a double negative gate, with essential repressive effects of unliganded TR.
DOI: https://doi.org/10.7554/eLife.45181.032

binucleated. TH similarly promotes the melanization of melanoblasts ex vivo and a cessation of proliferative activity in vivo (*McMenamin et al., 2014*). Our findings are broadly consistent with a role for TH in balancing proliferation and differentiation (*Brent, 2012*) and may be of clinical relevance, as human melanoma is associated with hypothyroidism and recurrent TH pathway mutations (*Ellerhorst et al., 2003*; *Shah et al., 2006*; *Sisley et al., 1993*). We suggest a model in which TH normally curtails expansion of the adult melanophore population by ensuring that cells cease to divide in a timely manner; in hypothyroid fish, the inappropriate retention of an immature state allows continued growth of the melanophore population during these post-embryonic stages. Whether TH induces a true cellular senescence and proliferative arrest, or whether cells at an apparently terminal state of differentiation remain competent to divide in specific conditions, will be interesting to learn.

TH promoted the terminal differentiation of xanthophores, but in a manner distinct from melanophores. We found far fewer TH-dependent genes in xanthophores than melanophores, likely reflecting the different developmental histories of these cells. In contrast to adult melanophores that arise from a transit-amplifying progenitor, most adult xanthophores develop directly from EL xanthophores that lose their pigment and then regain color late in adult pattern formation (*McMenamin et al., 2014*; *Patterson et al., 2014*) when TH levels are rising (*Chang et al., 2012*). The yellow-orange color of xanthophores can depend on pteridine pigments, carotenoid pigments, or both (*Bagnara and Matsumoto, 2006*; *Granneman et al., 2017*; *Lister, 2019*; *Odenthal et al., 1996*; *Ziegler, 2003*). We showed that TH directly or indirectly regulates carotenoid-associated genes and carotenoid deposition, allowing cryptic xanthophores to reacquire visible pigmentation. TH did not similarly influence pteridine pathway genes. These observations suggest that TH mediates a transition from pteridine-dependent pigmentation at embryonic/early larval stages to

carotenoid-dependent pigmentation of the same cells in the adult. Consistent with the notion of TH-mediated pigment-type switching, TH-dependent *scarb1* was required for carotenoid accumulation during adult pattern formation, yet mutants lacked an embryonic/early larval xanthophore phenotype. Conversely, mutants with pteridine and color deficiencies in embryonic/early larval xanthophores have normally pigmented adult xanthophores (*Lister, 2019*; *Odenthal et al., 1996*). In xanthophores at post-embryonic stages, then, TH drives a state of terminal differentiation from a developmental program that is relatively more advanced than that of progenitor-derived melanophores. That cryptic xanthophores appear poised to re-differentiate likely explains the smaller proportion of genes that were TH-dependent in these cells as compared to melanophores.

Finally, our study provides clues to likely roles for TRs during adult pigment pattern formation. TR mutants lacked overt pigmentation defects yet allowed for rescues of both melanophore and xanthophore defects in hypothyroid fish, suggesting that unliganded TRs normally repress maturation of these lineages. Loss of TRs similarly allows the survival of congenitally hypothyroid mice (*Flamant et al., 2002*; *Flamant and Samarut, 2003*). TRs may therefore prevent the inappropriate activation of gene expression programs required for lineage maturation when TH levels are low, as is thought to occur during amphibian metamorphosis (*Choi et al., 2015*; *Shi, 2013*). Although we detected pigment cell expression of each TR locus, our analyses cannot indicate whether TH acts directly on pigment cells through TR activities that are autonomous to these lineages. A plausible alternative would be that TH acts on stromal or other cell types in which TRs might be expressed and might exert similarly repressive effects when unliganded. Indeed, stromal cells of the hypodermis (*Lang et al., 2009*) and also iridophores (*Frohnhöfer et al., 2013*; *Patterson and Parichy, 2013*) regulate melanophore and xanthophore numbers during adult pigment pattern formation, and we observed striking differences in iridophore maturation depending on TH status. On-going efforts seek to distinguish between these possibilities. Results of the current study, however, represent a useful first step in understanding how globally available signals can control fine-grained patterning of cells within this complex adult trait.

# Materials and methods

**Key resources table**

| Reagent type (species) or resource | Designation | Source or reference | Identifiers | Additional information |
|---|---|---|---|---|
| Gene (*Danio rerio*) | bco1 | this paper | NCBI_Reference_Sequence: NM_001328495.1 | Amplified from cDNA |
| Gene (*Danio rerio*) | bco2b | this paper | NCBI_Reference_Sequence: NM_001040312.1 | Amplified from cDNA |
| Gene (*Danio rerio*) | bscl2l | this paper | NCBI_Reference_Sequence: NM_001013553.2 | Amplified from cDNA |
| Gene (*Danio rerio*) | slc2a11b | this paper | NCBI_Reference_Sequence: NM_001114430.1 | Amplified from cDNA |
| Gene (*Danio rerio*) | slc22a7a | this paper | NCBI_Reference_Sequence: M_001083861.1 | Amplified from cDNA |
| Gene (*Danio rerio*) | wu:fc46h12 | this paper | NCBI_Reference_Sequence: NM_001291347.1 | Amplified from cDNA |
| Gene (*Danio rerio*) | alx4a | this paper | NCBI_Reference_Sequence: XM_001340930 | Amplified from cDNA |
| Gene (*Danio rerio*) | alx4b | this paper | NCBI_Reference_Sequence: NM_001310078.1 | Amplified from cDNA |

*Continued on next page*

*Continued*

| Reagent type (species) or resource | Designation | Source or reference | Identifiers | Additional information |
|---|---|---|---|---|
| Gene (*Danio rerio*) | crip2 | this paper | NCBI_Reference_ Sequence: NM_001005968.1 | Amplified from cDNA |
| Gene (*Danio rerio*) | defbl1 | this paper | NCBI_Reference_ Sequence: NM_001081553.1 | Amplified from cDNA |
| Strain, strain background (*Danio rerio*) | Tg (tg:nVenus-v2a-nfnB) | PMID:25170046 | NA | NA |
| Strain, strain background (*Danio rerio*) | WT(ABb) | PMID:23737760 | NA | NA |
| Strain, strain background (*Danio rerio*) | Tg(aox5:palm EGFP)wp.rt22 | PMID:25170046 | NA | NA |
| Strain, strain background (*Danio rerio*) | Tg(tyrp1b:palm-mCherry)wp.rt11 | PMID:25170046 | NA | NA |
| Strain, strain background (*Danio rerio*) | Tg(−28.5Sox 10:Cre)zf384 | Gift. PMID:23155370 | NA | NA |
| Strain, strain background (*Danio rerio*) | Tg(−3.5ubi: loxP-eGFP-loxP -mCherry)cz1701 | Gift. PMID:21138979 | NA | NA |
| Strain, strain background (*Danio rerio*) | Tg(tuba8l3: nEosFP)vp.rt17 | this paper | NA | NA |
| Strain, strain background (*Danio rerio*) | thraa$^{vp33rc1}$ | this paper | NA | NA |
| Strain, strain background (*Danio rerio*) | thrab$^{vp31rc1}$ | this paper | NA | NA |
| Strain, strain background (*Danio rerio*) | thrb$^{vp34rc1}$ | this paper | NA | NA |
| Strain, strain background (*Danio rerio*) | scarb1$^{vp32rc1}$ | this paper | NA | NA |
| Strain, strain background (*Danio rerio*) | tyr$^{vp35rc1}$ | this paper | NA | NA |
| Antibody | anti-Dig-AP, sheep polyclonal Fab fragments | Millipore-Sigma | SKU_millipore-sigma:11093274910 | 1:5000 overnight at 4°C |
| Antibody | anti-GFP rabbit polyclonal antibody | Thermo Fisher | CatalogNo_A-11122 | 1:1000 overnight at 4°C |
| Commercial assay or kit | Lysotracker Far Red | Thermo Fisher | CatalogNo_L12492 | 75 nM |
| Commercial assay or kit | Vybrant DyeCycle Violet stain | Thermo Fisher | CatalogNo_V35003 | 5 µM |
| Commercial assay or kit | Senescence associated β-Galactosidase Staining Kit | Cell Signaling Technologies | CatalogNo_9860 | NA |

*Continued on next page*

*Continued*

| Reagent type (species) or resource | Designation | Source or reference | Identifiers | Additional information |
|---|---|---|---|---|
| Chemical compound, drug | Metronidazole | Acros Organics | CatalogNo_210341000 | NA |
| Chemical compound, drug | Oil Red O | Millipore-Sigma | CatalogNo_3125–12 | 5 mM |
| Software, algorithm | Cellranger | 10X Genomics | v2.0.2 | |
| Software, algorithm | Monocle | NA | v2.9.0 and v2.99.1 | https://github.com/cole-trapnell-lab/monocle-release.git |

## Staging, rearing and stocks

Staging followed (*Parichy et al., 2009*) and fish were maintained at ~28.5°C under 14:10 light:dark cycles. All thyroid-ablated (Mtz-treated) and control (DMSO-treated) *Tg(tg:nVenus-v2a-nfnB)* fish were kept under TH-free conditions and were fed only *Artemia*, rotifers enriched with TH-free Algamac (Aquafauna), and bloodworms. Fish stocks used were: wild-type AB^wp^ or its derivative WT(ABb) (*Eom et al., 2015*); *Tg(tg:nVenus-v2a-nfnB)^wp.rt8^*, *Tg(aox5:palmEGFP)^wp.rt22^*, *Tg(tyrp1b:palm-mCherry)^wp.rt11^* (*McMenamin et al., 2014*); *csf1ra^j4blue^* (*Parichy et al., 1999*); *Tg(−28.5Sox10:Cre)^zf384^* (*Kague et al., 2012*); *Tg(−3.5ubi:loxP-EGFP-loxP-mCherry)^cz1701^* (*Mosimann et al., 2011*); *tuba8l3:nEosFP^vp.rt17^*, *thrab^vp31rc1^*, *thraa^vp33rc1^*, *thrb^vp34rc1^*, *scarb1^vp32rc1^* and *tyr^vp35rc1^* (this study). Mutants and transgenic lines were maintained in the WT(ABb) genetic background. Fish were anesthetized prior to imaging with MS222 and euthanized by overdose of MS222. All procedures involving live animals followed federal, state and local guidelines for humane treatment and protocols approved by Institutional Animal Care and Use Committees of University of Virginia and University of Washington.

## Nitroreductase-mediated cell ablation

To ablate thyroid follicles of *Tg(tg:nVenus-2a-nfnB)*, we incubated 4-day post-fertilization (dpf) larvae for 8 hr in 10 mM Mtz with 1% DMSO in E3 media, with control larvae incubated in 1% DMSO in E3 media. For all thyroid ablations, treated individuals were assessed for loss of nuclear-localizing Venus (nVenus) the following day. Ablated thyroid glands fail to regenerate (*McMenamin et al., 2014*) and absence of regeneration in this study was confirmed by continued absence of nVenus expression.

## Mutant and transgenic line production

For CRISPR/Cas9 mutagenesis, one-cell stage embryos were injected with 200 ng/µl sgRNAs and 500 ng/µl Cas9 protein (PNA Bio) using standard procedures (*Shah et al., 2015*). Guides were tested for mutagenicity by Sanger sequencing and injected fish were reared through adult stages at which time they were crossed to *Tg(tg:nVenus-v2a-nfnB)* to generate heterozygous F1s from which single allele strains were recovered. CRISPR gRNA targets (excluding protospacer adjacent motif) are included in *Supplementary file 2—Table 7*. Mutant alleles of *scarb1* and TR loci are provided in *Figure 6—figure supplement 3* and *Figure 7—figure supplement 1*, respectively. The melanin free *tyr^vp.r34c1^* allele generated for analyses of melanophore lysosomal content exhibits a four nucleotide deletion beginning at position 212 that leads to novel amino acids and a premature stop codon (H71QEWTIESDGL*).

For F0 *thrb* mutagenesis analysis in the *thraa; thrab* mutant background, chemically synthesized Alt-R CRISPR-Cas9 sgRNAs targeting the *thrb* site and Cas9 protein (Alt-R S.p. Cas9 nuclease, v.3) were obtained from Integrated DNA Technologies (IDT). RNPs were prepared as recommended and ~1 nl was injected into the cytoplasm of one-cell stage embryos.

To label nuclei of adult melanophores, BAC CH73-199E17 containing the *puma* gene *tuba8l3* (*Larson et al., 2010*) was recombineered to contain nuclear-localizing photoconvertible fluorophore EosFP using standard methods (*Sharan et al., 2009*; *Suster et al., 2011*).

## Imaging

Images were acquired on: Zeiss AxioObserver inverted microscopes equipped with Axiocam HR or Axiocam 506 color cameras; a Zeiss AxioObserver inverted microscope equipped with CSU-X1 laser spinning disk (Yokogawa) and Orca Flash 4.0 camera (Hamamatsu Photonics); or a Zeiss LSM 880 scanning laser confocal microscope with Fast Airyscan and GaAsP detectors. Images were corrected for color balance and adjusted for display levels as necessary with conditions within analyses treated identically.

## Cell counts

Melanophores and xanthophores were counted within regions defined dorsally and ventrally by the margins of the primary stripes, anteriorly by the anterior margin of the dorsal fin, and posteriorly by five myotomes from the start. Only hypodermal melanophores were included in analysis; dorsal melanophores and those in scales were excluded. Mature xanthophores were counted by the presence of autofluorescent carotenoid with associated yellow pigment. Cell counts were made using ImageJ. Individual genotypes of fish assessed were confirmed using PCR or Sanger sequencing.

## In situ hybridization

In situ hybridization (ISH) probes and tissue were prepared as described (*Quigley et al., 2004*). Probes were hybridized for 24 hr at 66°C. Post-hybridization washes were performed using a BioLane HTI 16Vx (Intavis Bioanalytical Instruments), with the following parameters: 2x SSCT 3 × 5 min, 11 × 10 min at 66°C; 0.2x SSCT 10 × 10 min; blocking solution [5% normal goat serum (Invitrogen), 2 mg/mL BSA (RPI) in PBST] for 24 hr at 4°C; anti-Dig-AP, Fab fragments (1:5000 in blocking solution, Millipore-Sigma) for 24 hr at 4°C; PBST 59 × 20 min. AP staining was performed as described (*Quigley et al., 2004*).

## Pigment analyses

Xanthophore pigments were examined by imaging autofluorescence in eGFP and DAPI spectral ranges for carotenoids and pteridines, respectively. For imaging pteridines, fish were euthanized and treated with dilute ammonia to induce autofluorescence (*Odenthal et al., 1996*).

For analyses of carotenoid contents by HPLC we pooled three skin samples from each genotype and condition (Mtz-treated or control) into two separate samples. We homogenized the tissue in a glass dounce homogenizer with 1 ml of 0.9% sodium chloride and quantified the protein content of each sample with a bicinchoninic acid (BCA) assay (23250, Thermo). We then extracted carotenoids by combining the homogenates with 1 ml methanol, 2 ml distilled water, and 2 ml of hexane:*tert*-methyl butyl ether (1:1 vol:vol), separated the fractions by centrifuging, collected the upper solvent fraction, and dried it under a stream of nitrogen. We saponified these extracts with 0.2 M NaOH in methanol at room temperature for 4 hr following the protocol described in *Toomey and McGraw (2007)*. We extracted the saponified carotenoids from this solution with 2 ml of hexane:*tert*-methyl butyl ether (1:1 vol:vol) and dried the solvent fraction under a stream of nitrogen. We resuspended the saponified extracts in 120 µl of methanol:acetonitrile 1:1 (vol:vol) and injected 100 µl of this suspension into an Agilent 1100 series HPLC fitted with a YMC carotenoid 5.0 µm column (4.6 mm ×250 mm, YMC). We separated the pigments with a gradient mobile phase of acetonitrile:methanol:dichloromethane (44:44:12) (vol:vol:vol) through 11 min, a ramp up to acetonitrile:methanol:dichloromethane (35:35:30) for 11–21 min and isocratic conditions through 35 min. The column was warmed to 30°C, and mobile phase was pumped at a rate of 1.2 ml min$^{-1}$ throughout the run. We monitored the samples with a photodiode array detector at 400, 445, and 480 nm, and carotenoids were identified and quantified by comparison to authentic standards (a gift of DSM Nutritional Products, Heerlen, The Netherlands). Analyses of 5 dpf wild-type and *csf1ra* mutants used only larval heads where xanthophores are abundant in the wild type; other procedures were the same as for later stages.

## Immunohistochemistry and Oil-red-O staining

Skins of *Tg(aox5:palmEGFP)* euthyroid and hypothyroid zebrafish (8.6–10.4 SSL) were dissociated and plated at low density in L-15 medium (serum free) on collagen-coated, glass bottom dishes (Mattek) for 5 hr. Cells were then fixed with freshly prepared 4% PFA for 15 m, rinsed with PBST

(0.1%), blocked (5% goat serum, 1% BSA, 1X PBS), then incubated at 4°C overnight with rabbit anti-GFP primary antibody (ThermoFisher). Stained cells were rinsed 3X with 1X PBS and fixed again with 4% PFA for 30 min. Cells were then rinsed twice with ddH2O, washed with 60% isopropanol for 5 min, and then dried completely. Cells were incubated with filtered, Oil Red O solution (5 mM in 60% isopropanol) for 10 min, and rinsed 4X with ddH20 before imaging (*Koopman et al., 2001*). All GFP + cells were imaged across two plates per condition and were scored for presence or absence of red staining.

## Melanophore maturation assays

For assaying senescence of melanophores ex vivo, skins from euthyroid and hypothyroid fish (*n* = 3 each, 11 SSL) were cleared of scales, dissociated and plated on glass-bottom, collagen coated dishes (MatTek) in L-15 medium (Gibco) and incubated overnight at 28°C. Cells were then rinsed with dPBS, fixed with 4% PFA and stained using a Senescence β-Galactosidase Staining Kit (Cell Signaling Technologies, cat. #9860) according to manufacturer's instructions (*Ceol et al., 2011*; *Dimri et al., 1995*). Staining was carried out for 48 hr at pH six prior to imaging.

To assay cell state as measured by lysosomal content (*Kurz et al., 2000*; *Lee et al., 2006*) of melanophores by FACS, skins from euthyroid and hypothyroid *Tg(tyrp1b:palm-mCherry; tuba8l3:nEOS)*, *tyr* fish lacking melanin (*n* = 12 each) were dissociated and resuspended 1% BSA/5% FBS/dPBS. Cells were incubated for 1 hr with Lysotracker (75 nM) (ThermoFisher, L12492) and Vybrant DyeCycle Violet stain (5 µM) (ThermoFisher, V35003) shaking at 500 rpm, 28°C. Without washing, cells were FAC sorted. Single transgene controls and wild-type cells were used to adjust voltage and gating. Prior to analysis of fluorescence levels, single cells were isolated by sequentially gating cells according to their SSC-A vs. FSC-A, FSC-H vs FSC-W and SSC-H vs SSC-W profiles according to standard flow cytometry practices. Intact live cells were then isolated by excluding cells with low levels of DyeCycle violet staining (DAPI-A). As expected these cells express a wide range of our *tuba8l3:nlsEosFP* transgene as determined by levels of green fluorescence (FITC-A). Melanophores were isolated by identifying cells with high fluorescence in the FITC-A and mCherry-A channels which describe expression of the *tuba8l3:nlsEosFP* and *tyrp1b:palm-mCherry* transgenes. Lastly, lysosomal content of melanophores was determined by the median fluorescence intensity of the lysosomal marker, Lysotracker Deep Red (APC-A). The data were collected on a FACS ARIA using FACSDiva version eight software (BD Biosciences) and analyzed using FlowJo v10.

Melanin content was measured from brightfield images in Fiji. All image quantifications were performed using the base processing and analysis functions in ImageJ. Images were aligned and centered on the horizontal myoseptum and cropped to 2500 × 1500 pixels around dorsal and ventral stripes. Images were segmented based on red channel intensity using 'Auto Local Threshold' with parameters 'method = Sauvola radius = 50'. To account for close or overlapping melanophores, particles were further segmented using watershed segmentation. Particles larger than 25 pixels and not touching an edge were used for subsequent analyses.

## Transmission electron microscopy

Fish were euthanized then fixed in sodium cacodylate buffered 4% glutaraldehyde overnight at 4°C. Trunk regions were dissected then tissue stained in 2% osmium tetroxide for 30 min, washed, and then stained in 1% uranyl acetate overnight at 4°C. Samples were dehydrated with a graded ethanol series then infiltrated with a 1:1 propylene oxide:Durcupan resin for 2 hr followed by fresh Durcupan resin overnight and flat embedded prior to polymerization. Blocks were thin sectioned on a Leica EM UC7 and sections imaged on a JEOL 1230 transmission electron microscope.

## Tissue dissociations and FACS

Trunks or skins of staged, post-embryonic zebrafish (7.2–11.0 SSL) were dissected (*n* = 8 per replicate) and enzymatically dissociated with Liberase (Sigma-Aldrich cat. 5401119001, 0.25 mg/mL in dPBS) at 25°C for 15 min followed by manual trituration with a flame polished glass pipette for 5 min. Cell suspensions were then filtered through a 70 µm Nylon cell strainer to obtain a single cell suspension. Liberated cells were re-suspended in 1% BSA/5% FBS in dPBS and DAPI (0.1 µg/mL, 15 min) before FACS purification. All plastic and glass surfaces of cell contact were coated with 1% BSA in dPBS before to use. Prior to sorting for fluorescence levels, single cells were isolated by

sequentially gating cells according to their SSC-A vs. FSC-A, FSC-H vs FSC-W and SSC-H vs SSC-W profiles according to standard flow cytometry practices. Cells with high levels of DAPI staining were excluded as dead or damaged. Cells from wild-type and *Tg(ubi:switch)* zebrafish without Cre were used as negative control to determine gates for detection of mCherry and GFP fluorescence, then cells from *Tg(sox10:Cre; ubi:switch)* zebrafish were purified according to these gates. NC-derived cells cells were isolated by identifying cells with high fluorescence in the mCherry-A channel which describes expression of the *ubi:loxP-EGFP-loxP-mCherry* transgene after permanent conversion to *ubi:mCherry* after exposure to *Sox10:Cre* (see *Figure 2—figure supplement 1C*). All samples were kept on ice except during Liberase incubation, and sorted chilled.

## RT-PCR
Skin tissue from stage-matched fish was dissociated as above and melanophores and xanthophores were FAC sorted for the presence *aox5:palmeGFP* or *tyrp1b:palm-mCherry*, respectively. RNA was extracted from pools of 1000 cells using the RNAqueous-Micro kit (Thermo Fisher, cat. AM1912). Full length cDNA was synthesized with Superscript III reverse transcriptase (Thermo Fisher, cat. #18080093). Amplifications were 40 cycles with Q5 DNA polymerase (NEB, M0492), 38 cycles at 94° C, 30 s; 67°C, 20 s; 72°C, 20 s. For primer sequences (*actb1*, *thraa*, *thrab*, *thrb*), see *Supplementary file 2—Table 7*.

## Single-cell collection, library construction and sequencing
Whole-trunks or skins were collected from stage-matched *Tg(tg:nVenus-2a-nfnB)* euthyroid and hypothyroid siblings, dissociated, and *sox10*:Cre:mCherry+ cells isolated by FACS.

We replicated the experiment three times. For each replicate, we collected cells from euthyroid and hypothyroid fish at 7.2 SSL, 8.6 SSL, and 9.6 SSL (mid-larval, 6–10 fish per stage, per replicate) and sorted equal numbers of mCherry+ cells from each group into a single sample. Cells were pelleted and resuspended in 0.04% ultrapure BSA (ThermoFisher Scientific). Representing a terminal stage of pigment pattern development, we also collected mCherry+ cells from one sample within each replicate of 11 SSL (juvenile, five fish per condition) euthyroid and hypothyroid fish. To capture cells representing the EL pigment pattern, we collected mCherry+ cells from five dpf larvae (50 fish). In each experiment, we ran parallel euthyroid and hypothyroid samples (fish were siblings). For each sample, we targeted 2000–4000 cells for capture using the Chromium platform (10X Genomics) with one lane per sample. Single-cell mRNA libraries were prepared using the single-cell 3' solution V2 kit (10X Genomics). Quality control and quantification assays were performed using a Qubit fluorometer (Thermo Fisher) and a D1000 Screentape Assay (Agilent). Libraries were sequenced on an Illumina NextSeq 500 using 75-cycle, high output kits (read 1: 26 cycles, i7 Index: eight cycles, read 2: 57 cycles). Each sample was sequenced to an average depth of 150 million total reads. This resulted in an average read depth of ~40,000 reads/cell after read-depth normalization.

## scRNA-Seq data processing
We found that for many genes, annotated 3' UTRs in the Ensembl 93 zebrafish reference transcriptome were shorter than true UTR lengths observed empirically in pileups of reads mapped to the genome. This led to genic reads being counted as intergenic. To correct for this bias in aligning reads to the transcriptome, we extended all 3' UTR annotations by 500 bp. In rare cases, UTR extension resulted in overlap with a neighboring gene and in these instances we manually truncated the extension to avoid such overlap. We built a custom zebrafish STAR genome index using gene annotations from Ensembl GRCz11 with extended 3' UTRs plus manually annotated entries for mCherry transcript, filtered for protein-coding genes (with Cell Ranger *mkgtf* and *mkref* options). Final cellular barcodes and UMIs were determined using Cell Ranger 2.0.2 (10X Genomics) and cells were filtered to include only high-quality cells. Cell Ranger defaults for selecting cell-associated barcodes versus barcodes associated with empty partitions were used. All samples were aggregated (using 10X Cell Ranger pipeline 'cellranger aggr' option), with intermediary depth normalization to generate a gene-barcode matrix containing ~25,000 barcoded cells and gene expression counts.

## UMAP visualization and clustering

We used Uniform Manifold Approximation and Projection (UMAP) (*McInnes et al., 2018*) to project cells in two or three dimensions and performed louvain clustering (*Blondel et al., 2008*) using the reduceDimension and clusterCells functions in Monocle (v.2.99.1) using default parameters (except for, reduceDimension: reduction_method = UMAP, metric = cosine, n_neighbors = 30, mid_dist = 0.5; clusterCells: res = 1e-3, k = 15). We assigned clusters to cell types based on the detection of published marker genes. Cells isolated from euthyroid and hypothyroid fish were combined to maintain consistency of analysis and for comparisons between groups. Batch correction methods were not used between the two groups or across samples because we did not observe sample-specific separation or clustering in UMAP space. Cells with more than 15,000 UMIs were discarded as possible doublets. All genes were given as input to Principal Components Analysis (PCA). The top 30 principal components (high-loading, based on the associated scree plot) were then used as input to UMAP for generating either 2D or 3D projections of the data. For, subclustering of pigment cell clusters (melanophores, iridophores, xanthophores, and pigment progenitors), we subsetted the data set and again applied UMAP dimensionality reduction and louvain clustering.

## Differential expression analysis to determine cell-type markers

To identify genes expressed cell-type specifically, we used the principalGraphTest function in Monocle3 (v.2.99.1) with default parameters (*Cao et al., 2019*). This function uses a spatial correlation analysis, the Moran's I test, to assess spatially restricted gene expression patterns in low dimensional space. We selected markers by optimizing for high specificity, expression levels and effect sizes within clusters (For extended list of cell-type-specific genes, see *Supplementary file 2—Table 1*).

## Trajectory analysis

The top 800 highly dispersed genes (*Supplementary file 2—Table 5*) within euthyroid pigment cells (melanophores, xanthophores, iridophores, and pigment progenitors) were chosen as feature genes to resolve pseudotemporal trajectories using the setOrderingFilter, reduceDimension, and orderCells functions in Monocle (v2.9.0) using default parameters with the exception of setting max_components = 3 and num_dim = 10 to generate the trajectory in 3D with the top 10 PCs (high-loading based on scree plot) during dimensionality reduction.

## Branched Expression Analysis Modeling (BEAM)

After running trajectory analysis on pigment cells, we used the BEAM function in Monocle (v.2.9.0) with default settings (except, branch_point = 3) to determine differentially expressed genes between trajectory branches. To generate the BEAM heatmap for the three pigment cell trajectory branches, we used the plot_multiple_branches_heatmap function with default settings (except assigning branch 1, 5, and six to iridophores, melanophores, and xanthophores, respectively; and num_clusters = 6). Genes were selected by significance levels for the three-branch BEAM analysis with additional significant genes added from the melanophore and iridophore two-branch analysis for more even distribution of genes across lineages ($q < 6.0E-11$ for all genes, except for *pax3a* (starred, $q = 0.03$) which is a positive indicator of early pseudotime for all lineages).

## Differential expression analysis across pseudotime

To determine differentially expressed genes over pseudotime that were TH-dependent, we filtered the data set for genes expressed in at least five cells and performed differential expression analysis using a full model of sm.ns(Pseudotime, df = 3)*condition and a reduced model of sm.ns(Pseudotime, df = 3).

## Development and analysis of pathway signature scores

Gene sets for signature scores were selected using gene ontology (terms and gene sets from zfin. org; cell-cycle, unfolded protein response, AP-1 transcription factor complex members) or manual curation based on literature when required (carotenoid, pteridine, melanin) (see *Supplementary file 2—Table 4*). Signature scores were calculated by generating z-scores (using scale()) of the mean of expression values (log transformed, size factor normalized) from genes in a given set.

## Statistics

Parametric, non-parametric and multiple logistic regression analyses were performed using JMP 14.0 (SAS Institute, Cary, NC) or *R* [version 3.5.0] (*R Development Core Team, 2017*). For parametric analyses, residuals were assessed for normality and homoscedasticity to meet model assumptions and no transformations were found to be warranted.

## Data availability

Data is available on GEO via accession GSE131136.

## Code availability

Monocle is available through GitHub (https://github.com/cole-trapnell-lab/monocle-release.git; *Trapnell, 2019*).

## Acknowledgements

For assistance we thank D Jackson, D Huang, E Bain, B McCluskey, DS Eom, D Raible, A Fulbright, A Leith, A Schwindling, T Linbo, D White, and E Parker. Funding: Supported by NIH R35 GM122471 (DMP); NIH T32 GM007067 (LMS); NIH P30 EY001730 (Core Grant for Vision Research); NIH DP2 HD088158 (CT); Paul G Allen Frontiers Group - Allen Discovery Center grant (CT); WM Keck Foundation Grant (CT); Alfred P Sloan Foundation Research Fellowship (CT); NIH EY024958, EY025196, and EY026672 (JCC).

## Additional information

### Funding

| Funder | Grant reference number | Author |
| --- | --- | --- |
| National Institute of General Medical Sciences | R35 GM122471 | David M Parichy |
| Eunice Kennedy Shriver National Institute of Child Health and Human Development | DP2 HD088158 | Cole Trapnell |
| National Eye Institute | EY024958 | Joseph C Corbo |
| W. M. Keck Foundation | | Cole Trapnell |
| Alfred P. Sloan Foundation | | Cole Trapnell |
| Paul G Allen Frontiers Group | | Cole Trapnell |
| National Eye Institute | EY025196 | Joseph C Corbo |
| National Eye Institute | EY026672 | Joseph C Corbo |
| National Institute of General Medical Sciences | T32 GM007067 | Lauren M Saunders |

The funders had no role in study design, data collection and interpretation, or the decision to submit the work for publication.

### Author contributions

Lauren M Saunders, Conceptualization, Resources, Data curation, Formal analysis, Funding acquisition, Validation, Investigation, Visualization, Methodology, Writing—original draft, Writing—review and editing; Abhishek K Mishra, Matthew B Toomey, Investigation; Andrew J Aman, Victor M Lewis, Investigation, Writing—review and editing; Jonathan S Packer, Formal analysis, Methodology, Writing—review and editing; Xiaojie Qiu, Methodology, Writing—review and editing; Jose L McFaline-Figueroa, Methodology, Writing—original draft; Joseph C Corbo, Supervision, Funding acquisition, Writing—review and editing; Cole Trapnell, Conceptualization, Resources, Software, Supervision, Funding acquisition, Methodology, Writing—original draft, Project administration, Writing—review and editing; David M Parichy, Conceptualization, Resources, Formal analysis,

Supervision, Funding acquisition, Visualization, Methodology, Writing—original draft, Project administration, Writing—review and editing

### Author ORCIDs

Lauren M Saunders [iD] https://orcid.org/0000-0003-4377-4252
Matthew B Toomey [iD] http://orcid.org/0000-0001-9184-197X
Joseph C Corbo [iD] https://orcid.org/0000-0002-9323-7140
David M Parichy [iD] https://orcid.org/0000-0003-2771-6095

### Ethics

Animal experimentation: This study was performed in strict accordance with the recommendations in the Guide for the Care and Use of Laboratory Animals of the National Institutes of Health. All of the animals were handled according to approved institutional animal care and use committee (IACUC) protocols (4170) of the University of Vriginia and (4094-01) of the University of Washington. For imaging and other procedures animals were anesthetized with MS222 or euthanized by overdose of MS222 and every effort was made to minimize suffering.

### Decision letter and Author response

Decision letter https://doi.org/10.7554/eLife.45181.039
Author response https://doi.org/10.7554/eLife.45181.040

## Additional files

### Supplementary files

• Supplementary file 1. Interactive three-dimensional UMAP representation of transcriptomic space. Cells are colored by type corresponding to *Figure 2A*.
DOI: https://doi.org/10.7554/eLife.45181.033

• Supplementary file 2. scRNA seq analyses.
DOI: https://doi.org/10.7554/eLife.45181.034

• Transparent reporting form
DOI: https://doi.org/10.7554/eLife.45181.035

### Data availability

Data deposited in GEO under accession code GSE131136. Additional data are provided as source data files.

The following dataset was generated:

| Author(s) | Year | Dataset title | Dataset URL | Database and Identifier |
|---|---|---|---|---|
| Saunders LM, Parichy DM, Trapnell C | 2019 | Thyroid hormone regulates distinct paths to maturation in pigment cell lineages | https://www.ncbi.nlm.nih.gov/geo/query/acc.cgi?acc=GSE131136 | NCBI Gene Expression Omnibus, GSE131136 |

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
