## [Decision Letter]

Thank you for submitting your article "Thyroid hormone regulates distinct paths to maturation in pigment cell lineages" for consideration by *eLife*. Your article has been reviewed by three peer reviewers, one of whom is a member of our Board of Reviewing Editors, and the evaluation has been overseen by Didier Stainier as the Senior Editor. The following individuals involved in review of your submission have agreed to reveal their identity: Craig Ceol (Reviewer #2); Ian J Jackson (Reviewer #3).

The reviewers have discussed the reviews with one another and the Reviewing Editor has drafted this decision to help you prepare a revised submission.

Summary:

This manuscript defines how thyroid hormone has opposite effects on the abundance of melanophores and xanthophores during adult zebrafish pigment pattern development. The authors originally hypothesized that the imbalance in pigment cell lineages found in hypothyroid zebrafish is caused, in one model, by an amplification of one cell type at the cost of the other during cell-fate specification. In an alternative model, they hypothesized that thyroid hormone has discordant affects on a particular cellular behavior in both lineages. Thus, a perturbation in thyroid hormone function would result in the observed melanophore excess and xanthophore deficit seen in hypothyroid fish. In order to test these hypotheses, the authors used single-cell sequencing to profile thousands of neural crest-derived cells and reconstructed pigment cell developmental trajectories in pseudotime. A comparison of the euthyroid and hypothyroid trajectories revealed a buildup of both immature melanophores and xanthophores, which was incongruent with their first two models and caused the authors to seek another explanation. Utilizing their scRNA-Seq data, they investigated differentially expressed genes in melanophores and found that maturation genes were expressed at lower levels in hypothyroid fish when compared to euthyroid fish. Based on this transcriptional finding, the authors found that some euthyroid melanophores have greater staining for senescence markers when compared to hypothyroid melanophores. In analyzing the xanthophore lineage, the authors found that genes involved with carotenoid pigment production are differentially expressed between hypothyroid and euthyroid xanthophores. In support of this transcriptional finding they utilized a carotenoid mutant to show that immature xanthophores are still present but unable to mature. The authors combined these analyses to construct a new model in which thyroid hormone drives maturation of both adult xanthophores and melanophores, but via independent mechanisms. More broadly, the study provides a wealth of new markers and data for identifying distinct populations of pigment cell classes, which are likely to be of use beyond the zebrafish alone. Overall, the study is well executed and well presented to a non-computational audience.

Essential revisions:

1) Pigment cell subtypes

– In Figure 3, could the authors better explain exactly what differentiates the subtype classifications, i.e. mel1 vs. mel2; is there a morphological difference between these cell types that can be seen by ISH? Is there any possibility that the mel1 vs. mel2 states stratifies against binuclearity?

2) Binucleate cell state

Several issues were raised around these claims:

– In Figure 3—figure supplement 1, it is shown that certain cell types have lower transcriptional activity (i.e. mel2, xan1). In the scRNA-Seq method, how do you account for transcript abundance in relation to the fact that some of the cells being sequenced are binucleate whereas some are mononuclear? This seems like it would confound the transcript counts if this is not accounted for. A better explanation of this or experiments to demonstrate this is not the case would be helpful.

– The relationship between binuclear status and senescence in melanophores is tenuous. Previous literature (including Usui et al., 2018) states that binuclear melanophores are frequently the result of failures in cell division. The authors should carefully clarify the relationship between melanophore maturation, senescence, and binuclear status.

– Understanding the mechanisms of how the cell transitions from uni- to binuclearity would be useful. Are there correlations from the scRNA data that might be candidate mechanisms by which this could occur? Even if correlative, it would be helpful for future studies in the field.

3) The TR experiments

– For the unliganded TR receptors to suppress gene expression, are there markers that mediate this repression that you could show occurs in the hypothyroid fish? This would add strength to the genetic argument that the unliganded state is the correct mechanism. Does the unliganded receptor repress the same genes that the receptor + TH activate?

– The authors should display phenotype data from a *thrab(lf);thraa(lf);thrb(lf)* triple mutant. The authors observe no overt pigment defects in *thraa, thrab*, and *thrb* single knockouts as well as in the *thraa(lf);thrab(lf)* double knockout, however, to completely conclude that TR mutants don't have a pigment defect the authors should also include the *thrb* null allele. It is possible that in the *thraa(lf);thrab(lf)* mutant doesn't have a phenotype because Thyroid Receptor Β is compensating for their loss and mediating thyroid signaling. This concern also applies to the modest rescue seen in Figure 7B-D. A triple mutant may yield a full rescue of the xanthophore population. Imaging and subsequent quantification of the triple mutant would strengthen the authors' interpretations.

4) Senescence

– The authors need to improve the evidence behind their claim that senescence is promoted by thyroid hormone during melanophore maturation. SA-β-Gal is the gold-standard assay for senescence, and while the authors perform SA-β-Gal on euthyroid and hyperthyroid melanophores, they do not quantify the results. Therefore, it is not clear whether the one cell shown represents any appreciable fraction of euthyroid melanophores. Lysotracker is used as a secondary assay to support this idea, but increased lysotracker staining is not a specific feature of senescent cells and is more associated with autophagy. The authors need to bolster their claim that thyroid hormone promotes senescence by quantifying SA-β-Gal and investigate other well-accepted ways to corroborate cellular senescence, e.g. p16 expression.

---

## [Author Response]

Essential revisions:1) Pigment cell subtypes– In Figure 3, could the authors better explain exactly what differentiates the subtype classifications, i.e. mel1 vs. mel2; is there a morphological difference between these cell types that can be seen by ISH? Is there any possibility that the mel1 vs. mel2 states stratifies against binuclearity?

The pigment cell subtypes were revealed by unsupervised clustering and we cannot say with complete certainty which anatomical or morphological states they represent. Preliminary histological assessments have not been informative largely because one cluster of each cell type (i.e., mel2 and xan1) had comparatively few genes expressed and many of these are widely expressed by other cell types, making ISH results difficult to interpret. For this reason, we are undertaking considerably larger scale scRNA-Seq and other analyses of these populations, but such efforts will take considerable time to reach completion. Nevertheless, our current analyses of expression levels, numbers of genes expressed, and AP-1 signatures, allow some inferences about these populations. We modified the main text to make these points, while remaining conservative in our assessment:

“These analyses revealed subsets of melanophores and xanthophores (Figure 3B), consistent with differences in states of differentiation and morphogenetic behaviors (Eom et al., 2015; Parichy et al., 2000b; Parichy and Spiewak, 2015). For example, cells of melanophore subcluster 2 exhibited low levels of transcriptional activity and expressed fewer genes, suggesting a more advanced state of differentiation, as compared to cells of melanophore 1 (Figure 3—figure supplement 1) Likewise cells of xanthophore 1 had fewer transcripts and expressed genes than cells of xanthophore 2, suggesting they may represent undifferentiated, cryptic xanthophores (McMenamin et al., 2014) and actively differentiating populations, respectively.”

We also revised the Figure 3—figure supplement 1 to include new analyses and modified its legend to better explain these points:

“(C) Melanophores and xanthophores in subclusters with lower total UMI counts (mel2, xan1) expressed fewer unique genes compared to cells in the other subcluster regardless of equivalent UMI counts, consistent with a more restricted gene expression profile of these populations (shaded areas indicate standard error bounds). (D) Differential expression of genes between pigment cell sub-clusters (xan1 vs. xan2; mel2 vs. mel1). These analyses revealed more genes, expressed at higher levels in xanthophore 2 compared to xanthophore 1, and in melanophore 1 compared to melanophore 2. These biases were consistent with xanthophore 2 and melanophore 1 representing more active cells, and xanthophore 1 and melanophore 2 representing less active cells. Genes compared were expressed by at least 5 cells in either cluster (log_2_ fold-change cutoff = 0.8, P < 1e^-3^; xanthophores = 160, 36; melanophores = 125, 39. (E) Melanophore and xanthophore subclusters were differentially distributed along trajectory branches in Figure 3D, consistent with xanthophore 2 and melanophore 2 representing cells at later stages of maturation as compared to xanthophore 1 and melanophore 1.”

We do not have information on whether the binucleate state is specifically associated with one or the other cluster. Although our analyses strongly suggest that binucleation is a property of terminally differentiated melanophores, making these cells likely to be associated with mel2, screening of potential markers from this data set and others has yet to be informative. Indeed, we strongly suspect that binucleate cells may be under-sampled in our scRNA-Seq data owing to their large size and increased likelihood of being lost during FACS isolation or 10X Chromium capture. Thus, we anticipate that the binucleate pool might well form a subset that we cannot yet distinguish within a broader “mel2” pool. We are undertaking additional scRNA-Seq and other analyses to address these possibilities, but these are long-term goals beyond the scope of the present study, which does not depend on this information for the conclusions we have reached.

2) Binucleate cell stateSeveral issues were raised around these claims:– In Figure 3—figure supplement 1, it is shown that certain cell types have lower transcriptional activity (i.e. mel2, xan1). In the scRNA-Seq method, how do you account for transcript abundance in relation to the fact that some of the cells being sequenced are binucleate whereas some are mononuclear? This seems like it would confound the transcript counts if this is not accounted for. A better explanation of this or experiments to demonstrate this is not the case would be helpful.

As noted above we cannot formally assign binucleate cells to one or the other melanophore subcluster and we do not yet have a way to normalize transcript counts relative to either number of nuclei or individual cell size. While we think this is an interesting enough issue to continue pursuing, we do not believe our conclusions in the present study depend on a complete explication of these phenomena or their interrelationships. Our major intent is to simply document that subsets of these cells can be identified using our bioinformatic methods. We are also not convinced a bias of the sort apparently envisioned would be at play anyway. True, one might anticipate cells of larger size or more nuclei having more transcriptional activity. Yet we found that binucleation increases with fish and melanophore age (see below and new panel Figure 5—figure supplement 1E) and cells having transcriptional characteristics and pseudotime trajectory states of more mature cells, i.e., mel2, actually have less transcriptional activity and a less diverse repertoire of expressed genes. So, we suspect binucleate cells are actually less active and, if anything, having two nuclei (of low transcriptional activity) would likely tend to obscure the genomic and transcriptomic phenotype of each nucleus on its own. We are working to address these issues with scRNA-Seq, assessments of chromatin state, and alternative cell capture methods, but these efforts will not bear fruit for some time and are not essential to the present study.

– The relationship between binuclear status and senescence in melanophores is tenuous. Previous literature (including Usui et al., 2018) states that binuclear melanophores are frequently the result of failures in cell division. The authors should carefully clarify the relationship between melanophore maturation, senescence, and binuclear status.

We thank the reviewers for this raising this point as we realize that several different threads were woven together too densely for the sake of brevity. We have extensively modified our presentation of melanophore phenotypes in subsection “TH promotes a melanophore maturation program” and Discussion section. We emphasize binucleation as just one of several indicators of cell state, and although multi-nucleation is associated with proliferative senescence in other systems (see references in text), we do not assume or conclude this to be the case here.

We additionally clarified the relationship of SA-β-Gal and lysosomal content assays (see references in text), and we avoided referring to these assays (or binucleation) as indicators of a senescent phenotype per se. We have also avoided the term proliferative “arrest” because we do not know whether cells with high lysosomal content or multiple nuclei are autonomously “arrested,” or might be competent to continue mitosis or cytokinesis under specific circumstances. In addition to our original analyses, we added:

– An assessment of binucleation incidence across stages in euthyroid fish (Figure 5—figure supplement 1E), demonstrating its higher prevalence in more mature melanophores.

– Lysotracker staining in vitro, showing that melanosomes are not labeled with this dye, despite having some similarities to lysosomes during their earlier biogenesis (Figure 5—figure supplement 1B).

– FACS assessment of euthyroid and hypothyroid differences in forward scatter, often associated with overall cell size (Figure 5—figure supplement 1D).

– An assessment of melanin content in vivo using measurements of contracted melanosome areas, showing that hypothyroid melanophores are under-melanized compared to euthyroid melanophores (Figure 5B,C).

All of these observations support our model that TH promotes melanophore maturation, regardless of proliferative senescence per se.

– Understanding the mechanisms of how the cell transitions from uni- to binuclearity would be useful. Are there correlations from the scRNA data that might be candidate mechanisms by which this could occur? Even if correlative, it would be helpful for future studies in the field.

We, too, want to understand these transitions but preliminary investigations focusing on obvious candidates have not been informative. We continue to pursue this issue outside the scope of the present study.

3) The TR experiments– For the unliganded TR receptors to suppress gene expression, are there markers that mediate this repression that you could show occurs in the hypothyroid fish? This would add strength to the genetic argument that the unliganded state is the correct mechanism. Does the unliganded receptor repress the same genes that the receptor + TH activate?

This is a complex issue (e.g., Grontved et al., 2015) and will likely take several years of additional work to address rigorously, particularly as it may require analyses at single cell level, methods for which remain in their infancy. Nor do we know of a priori targets that would allow us to test this idea at small scale, as there is little consensus among existing datasets and none that demonstrate direct targets in our several cell types of interest. Indeed, and as noted at the end of the Discussion section, we do not yet know if TR activities are autonomous to melanophores and xanthophores, or if these receptors act indirectly through other cell types. We are working to understand these issues and believe they will need a full treatment on their own, independent of the many new findings we present in this study.

– The authors should display phenotype data from a thrab(lf);thraa(lf);thrb(lf) triple mutant. The authors observe no overt pigment defects in thraa, thrab, and thrb single knockouts as well as in the thraa(lf);thrab(lf) double knockout, however, to completely conclude that TR mutants don't have a pigment defect the authors should also include the thrb null allele. It is possible that in the thraa(lf);thrab(lf) mutant doesn't have a phenotype because Thyroid Receptor Β is compensating for their loss and mediating thyroid signaling. This concern also applies to the modest rescue seen in Figure 7B-D. A triple mutant may yield a full rescue of the xanthophore population. Imaging and subsequent quantification of the triple mutant would strengthen the authors' interpretations.

We added data for triple mutant cells in Figure 7. When all three receptor genes are knocked-out in euthyroid fish, we find no significant differences in xanthophore or melanophore phenotypes compared to wild-type. But in hypothyroid mutant fish we observe rescues of xanthophore and melanophore numbers back to euthyroid wild-type or mutant levels. Because de novo generation of a stable triple mutant line with the thyroid-ablation transgene in its background would have taken ~2 years, we elected to use Alt-R CRISPR/Cas9 to generate somatic mutations for *thrb* in a *thraa; thrab* double mutant transgenic background. Rescue of the hypothyroid phenotype back to a euthyroid state, without other defects evident, provides strong evidence of reagent specificity in addition to the repressive effects of TRs.

4) Senescence– The authors need to improve the evidence behind their claim that senescence is promoted by thyroid hormone during melanophore maturation. SA-β-Gal is the gold-standard assay for senescence, and while the authors perform SA-β-Gal on euthyroid and hyperthyroid melanophores, they do not quantify the results. Therefore, it is not clear whether the one cell shown represents any appreciable fraction of euthyroid melanophores. Lysotracker is used as a secondary assay to support this idea, but increased lysotracker staining is not a specific feature of senescent cells and is more associated with autophagy. The authors need to bolster their claim that thyroid hormone promotes senescence by quantifying SA-β-Gal and investigate other well-accepted ways to corroborate cellular senescence, e.g. p16 expression.

As noted above, we modified this presentation extensively. In our hands SA-β-Gal staining is so variable and subjective within and between TH states that we do not feel comfortable trying to quantify its incidence. SA-β-Gal staining is due to lysosomal β-Gal activity in other cell types, and so we use Lysotracker to assess lysosomal content quantitatively by FACS. We present these observations in the context of maturation overall, and in conjunction with other indicators of melanophore state (gene expression profile, melanin content, inferred cell size, binucleation incidence). Whether this suite of characteristics exhibited by euthyroid melanophores, plus their cessation of proliferative activity (as shown by us in McMenamin et al., 2014) represents a classical state of cellular senescence as observed in cell lines or in response to oncogenic stimulation in vivo remains uncertain. We do not observe *cdkn2a/b (p16*) expression in our cells, and the extent of natural (as opposed to oncogene-induced) cellular senescence in zebrafish is unclear (e.g., Kishi et al., 2006). Our presentation of the phenotype acknowledges this uncertainty in the Results section and Discussion section.